# RePAIR: A Rule-based Process-Adaptive Reinforcement for Large Language Model Training

## Abstract

Although reinforcement learning (RL) has demonstrated promise in enhancing the reasoning capabilities of Large Language Models (LLMs), the difficulty of reward design has prohibited exploiting the full potential of RL. Previous methods mainly fall into two categories: training a reward model based on human preferences or designing verifiable outcome rewards. However, reward models often suffer from poor interpretability and require extensive annotation for effective training. Verifiable outcome rewards provide sparse signals only, which leads to an ambiguous credit assignment and low training efficiency in RL. These limitations necessitate rewards that provide more efficient, fine-grained supervision. In order to address these, we propose **R**ule-based **P**rocess-**A**dapt**I**ve **R**einforcement (RePAIR) that constructs adaptive verifiable process rewards through symbolic reasoning rules. These rules are automatically derived through the integration of common pattern mining and semantic summarization over the reasoning trajectories of LLMs. For stable training purposes, RePAIR defines a reward informativeness metric that dynamically adjusts the rule's weights based on policy updates. Extensive experiments across three reasoning tasks demonstrate that RePAIR achieves a 6.03% improvement on average and combines well with various advantage functions. Code and data will be available at https://anonymous.4open.science/r/RePAIR-8EFC.

## 1 Introduction

Reinforcement Learning (RL) has emerged as a promising paradigm for enhancing the reasoning capabilities of large language models (LLMs), particularly in tasks involving multi-step generation strategies Jaech et al. (2024); DeepSeek-AI et al. (2025) and alignment with human preferences Lin et al. (2025). Notably, the effectiveness of RL heavily depends on the reward design, which serves as the core feedback signal that guides model optimization Zhong et al. (2025). Different from traditional RL, where the environment is well-defined with clear structures and regularities, e.g., physical laws, and the consequences of agent's actions can be accurately evaluated Sutton & Barto (2018), when applying RL to LLMs, the conventional "simulatable environment" is replaced by a black-box generative system driven by an LLM Ouyang et al. (2022b). In this case, the state transition process is entirely driven by parameters within the LLM, which introduces a high degree of uncertainty and lacks clear structure or verifiable dynamic rules. As a result, designing effective reward functions becomes significantly more complex and challenging.

Most prevailing methods that apply RL paradigms for LLMs employ the black-box preference model Lin et al. (2025) or the outcome scoring model Bai et al. (2022); Wang et al. (2024) to construct reward signals. However, such reward models lack interpretability and fail to reveal the causality between the agent's action and the reward feedback, which are prone to policy drift and preference bias Gao et al. (2022); Lightman et al. (2023). Moreover, in order to collect adequate high-quality labels for reward model training, researchers either build complicated human annotation pipelines Lightman et al. (2023) or rely on estimation-based methods, which require approximate $10\times$ more rollouts for each step than sampling the response-level trajectories only Wang et al. (2023b); Kazemnejad et al. (2024). In order to cope with these problems, very recently, verifiable reward has been proposed to provide clear binary feedback through a rule-based reward function Lambert et al. (2024); DeepSeek-AI et al. (2025), which avoids subjective human assessments and complex reward models

training. However, the verifiable outcome rewards employed by industry-leading models DeepSeek-AI et al. (2025) suffer from the challenges of reward sparsity and credit assignment Leike et al. (2018), which fail to capture long-term dependencies and uncertainties in intermediate steps within LLM-generated sequences Cui et al. (2025).

In order to tackle these challenges, a *verifiable process reward* is desired, where fine-grained interpretable feedback to intermediate reasoning steps Setlur et al. (2024) can be provided. However, it is not trivial to define verifiable process rewards for LLM tasks as follows: (1) Ambiguity of task goals: since the goals in LLM tasks are often ambiguous, the process reward criteria lack clear quantitative boundaries, which are highly dependent on human subjective judgment. (2) High-dimensional and unstructured action space: the output of LLMs is a high-dimensional sequence Ouyang et al. (2022b), and the action space is the entire vocabulary, up to tens of thousands or even hundreds of thousands of tokens, which implicitly encodes syntactic, semantic, and logical contextual information. As a result, verifying intermediate steps becomes extremely complex, which makes it hard to design reward functions that are both objective and consistent. In contrast, traditional RL tasks benefit from low-dimensional and discrete spaces, where such complexity does not arise. (3) Task-specific variability: different tasks have their own specific reasoning logic and semantic structure, which makes it hard to design a universal process reward function Chung et al. (2024). For each new task, it requires a costly and unscalable redesign by domain experts. (4) Adaptivity to model's update: an ideal reward must be dynamically adaptive, as a static reward eventually leads to overoptimization or reward hacking Gao et al. (2022) due to distribution shift. Moreover, the variability in LLM outputs further demands that rewards adapt to policy and environment shifts to ensure generalization and robustness

We propose a rule-based approach (RePAIR) to construct verifiable process rewards, which provides fine-grained, generalizable, and adaptive supervision for reinforcement learning in LLMs. RePAIR treats symbolic reasoning rules, extracted from reasoning trajectories, as the physical laws of the LLM-generated reasoning environment. These rules formalize reasoning patterns as computable logical expressions, thereby providing verifiable and structured constraints in the uncertain and high-dimensional generation space of LLMs. As for the automatic extraction of these rules, RePAIR first converts natural language reasoning trajectories into graphs, which facilitates the identification of common reasoning patterns. These patterns, combined with the semantic features of the reasoning trajectories, are then formalized into symbolic reasoning rules via an LLM. Moreover, for the purpose of efficient and stable policy learning, it dynamically adjusts rule weights during training. Meanwhile, our research focuses on smaller-parameter LLMs (e.g., 0.5B, 1.5B), which are particularly suitable for edge deployment, personalization, and privacy-preserving applications. Under limited computational budgets, these models offer an efficient balance between performance and resource consumption. Our main contributions are summarized as follows.

- **Symbolic reasoning rules.** We automatically extract symbolic reasoning rules from LLM-generated trajectories, which formalize common reasoning patterns as a computable function to provide a verifiable and interpretable basis for process supervision.

- **Adaptive and verifiable process rewards.** We transform symbolic reasoning rules into verifiable scalar signals and dynamically adjust rules' weights based on a reward informativeness metric, which enables adaptive and fine-grained reward shaping during learning.

- **Experimental results.** Extensive experiments on three tasks demonstrate the following: (1) RePAIR achieves a 6.03% performance improvement on average; (2) RePAIR is an algorithm-agnostic and universally applicable enhancement module for almost any RL algorithm in LLM training and removes the need for task-specific reward design; and (3) RePAIR enhances the model's ability to generalize beyond the training distribution.

## 2    RELATED WORK

### 2.1    RL FOR LLM REASONING

Reinforcement Learning plays a critical role in enhancing the instruction-following capabilities of LLMs through three representative paradigms: (1) *Reinforcement Learning from Human Feedback (RLHF)*: RLHF employs human-annotated preferences to train a reward model, which then guides policy optimization Ouyang et al. (2022a); Wang et al. (2024). Despite its effectiveness in alignment,

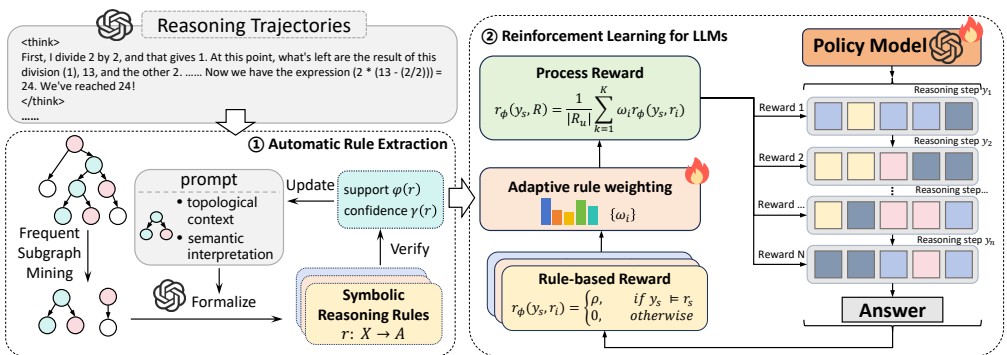

Figure 1: Overview of the RePAIR framework.

RLHF is constrained by annotation cost and the lack of interpretability. (2) *Reinforcement Learning from AI Feedback (RLAIF)*: RLAIF replaces human annotators with LLMs to automate feedback collection Kim et al. (2023) with more scalability. However, AI-generated preferences often reflect model biases and lack verifiability, which potentially reinforces errors during fine-tuning. (3) *Reinforcement Learning with Verifiable Rewards (RLVR)*: RLVR, inspired by DeepSeek Math/R1 DeepSeek-AI et al. (2025); Shao et al. (2024), is formally introduced in TÜLU 3 Lambert et al. (2024) as a framework that uses verifiable reward functions to automatically evaluate the correctness of a model's outputs via deterministic rules and provides binary reward signals. However, it relies on high-quality, verifiable datasets with ground-truth, which limits its applicability. Our method advances the RLVR paradigm beyond its original scope: instead of relying on manually designed reward functions grounded in expert-verified labels, we automatically extract symbolic rules from LLM-generated reasoning trajectories and integrate them into the reinforcement learning process as verifiable process rewards.

## 2.2 REWARD MODELS FOR LLM TRAINING

From the perspective of reward design, there are two main approaches distinguished by their granularity: Outcome Reward Models (ORM) DeepSeek-AI et al. (2025); Shao et al. (2024) and Process Reward Models (PRM) Luo et al. (2024); Zhang et al. (2024). ORM assigns rewards based on the final output labels, which suffers from delayed feedback and the credit assignment problem Yang et al. (2024b); Liu et al. (2024). In contrast, PRM evaluates intermediate reasoning steps to provide more fine-grained supervision. One of the most critical challenges in PRM is reward hacking, where models exploit superficial signals rather than truly following the intended reasoning trajectory Wang et al. (2023a). Furthermore, training PRM requires expensive human annotation Uesato et al. (2022), which makes large-scale implementation impractical. Different from previous methods, we leverage rules to provide process supervision without the requirement of extensive annotation and extra reward model training cost. In addition, our rule-based rewards adapt dynamically during training to better align LLM behavior with target reasoning patterns and mitigate reward hacking.

## 3 PRELIMINARIES

Reinforcement Learning aims to learn an optimal policy $\pi_\theta$ that maximizes the expected cumulative reward, namely return, when interacting with an environment. In the context of autoregressive language modeling, the state at step $t$ is the concatenation of input $\mathbf{x}$ and current response $o_{<t}$, and the action is the $t$-th token or step $y_t$. As a fundamental algorithm that optimizes the learning objective, policy gradient method focuses on the advantage function $A_t$ which quantifies how much better an action is compared to alternatives in a given state:

$$\nabla_\theta J(\theta) = \mathbb{E}_{\mathbf{x}\sim\mathcal{D},o\sim\pi_\theta}\left[\sum_{t=0}^{T}\nabla_\theta \log \pi_\theta(y_t|\mathbf{x}, o_{<t})A_t\right], \tag{1}$$

where $(\mathbf{x}, o)$ represents a pair of input and output. In practice, the advantage function is implemented as cumulative discounted rewards subtracting a baseline $A_t = \sum_{s=t}^{T}\gamma^{s-t}r(y_s) - b$, where $\gamma \in [0, 1]$

is a discount factor that optionally decays future rewards, and $r(y_s)$ is the reward provided by the environment at time step $s$ with $\mathbf{x}$ and $o_{<s}$ omitted in conditions.

# 4 RePAIR FRAMEWORK

In this section, we introduce the RePAIR framework, as shown in Figure 1, which consists of two stages: (1) automatic rule extraction from LLM-generated reasoning trajectories and (2) reinforcement learning with adaptive, verifiable process rewards constructed by these rules.

## 4.1 AUTOMATIC RULE EXTRACTION

The goal of automatic rule extraction is to derive symbolic reasoning rules from LLM-generated trajectories by identifying common reasoning patterns and abstracting their semantic features. It consists of the following two steps.

Frequent subgraph mining (FSM) is a graph-pattern discovery technique that extracts substructures which recur across a collection of graphs above a minimum-support threshold.

### 4.1.1 FREQUENT SUBGRAPH MINING

In the first step, we perform frequent subgraph mining (FSM) on the reasoning trajectories $\mathcal{T}$ to capture latent reasoning patterns. FSM is a widely used graph-based data mining method Khan et al. (2010); Yan & Han (2003) that aims to extract substructures which recur across a set of graphs above a minimum-support threshold. Specifically, for each task, we collect multiple reasoning trajectories and divide them into successful and failed trajectories based on their correctness. Each trajectory set is then modeled as a graph $G = (V, E)$, where nodes $V$ represent intermediate reasoning steps and edges $E$ denote dependencies among these steps. In order to identify common reasoning patterns, we transform each reasoning step into a vector-based semantic representation. Specifically, given some problems and their solutions, we prompt an LLM to summarize key semantic features, which are then encoded into a structured feature vector. Each reasoning step in the trajectory is thus mapped into an embedding space where semantically similar steps share aligned representations. These vector-based labels serve as semantic labels in the reasoning graph. An illustrative example is shown in Figure 2. Based on these labeled graphs, we apply the frequent subgraph mining method, GRAMI Elseidy et al. (2014), to extract a set of subgraphs $\mathcal{S}$ that appear at least $\sigma$ times.

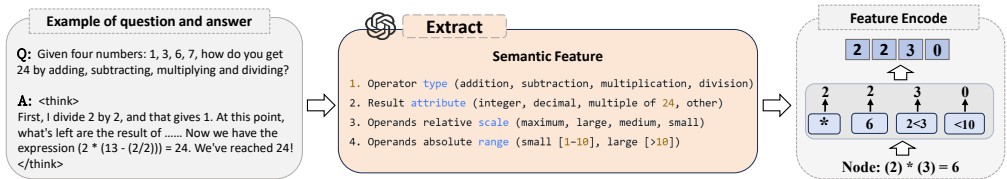

Figure 2: Example of semantic feature extraction and vector label construction on Game of 24 task.

### 4.1.2 RULE FORMALIZATION

In the second step, we formalize the discovered frequent subgraphs $\mathcal{S}$ into symbolic reasoning rules by prompting an LLM with structured descriptions of the nodes in each subgraph. Specifically, we construct a prompt $\pi = \text{Prompt}(\mathcal{S}, \text{desc}(v) \mid v \in V)$, where $\text{desc}(v)$ includes both the semantic interpretation of the node label and its topological context. The semantic interpretation is obtained from the statistical distribution of attribute values. Based on this prompt $\pi$, the LLM is instructed to generate specific, executable rules. We refer to them as symbolic reasoning rules expressed in **first-order logic expressions**:

$$r : X \to A, \tag{2}$$

where $X$ is a conjunction of predicates that describe the state of the reasoning step, and $A$ is a predicate that represents the corresponding action the model should take in this step. Each predicate is a Boolean function defined over the semantic attributes of the reasoning steps. We refer to $X$ as the precondition of $r$ and $A$ as the consequence of $r$. Note that $A$ can be an empty set, which means the rule encodes only state constraints without prescribing a specific action.

**Definition 1 (Rule Matching)** *Given a reasoning step instance $y_s$ and a rule $r : X \rightarrow A$, we say that $y_s$ matches $r$, denoted as $y_s \vDash r$, if $y_s$ simultaneously satisfies the rule's precondition $X$ and consequence $A$. More generally, we write $y_s \vDash X$ if $y_s$ satisfies only the precondition part $X$.*

Based on Definition 1, we introduce each rule's support and confidence over reasoning trajectories $\mathcal{T}$ Agrawal et al. (1993). For a rule $r : X \rightarrow A$, its support $\varphi(r)$ and confidence $\gamma(r)$ is defined as:

$$\varphi(r) = \frac{|\{T_i \mid y_s \vDash X \wedge A, y_s \in T_i\}|}{|\mathcal{T}|}, \quad \gamma(r) = \frac{|\{T_i \mid y_s \vDash X \wedge A, y_s \in T_i\}|}{|\{T_i \mid y_s \vDash X, y_s \in T_i\}|} \tag{3}$$

where $T_i \in \mathcal{T}$. Support measures the coverage of a rule across the reasoning trajectories, while confidence reflects the reliability of the rule's conclusion given that its precondition holds. For each symbolic reasoning rule generated by the LLM, we evaluate these metrics to verify its validity and robustness, ensuring that they capture genuine reasoning patterns rather than spurious correlations or hallucinations produced by the LLM. Only rules with sufficiently high support and confidence over $\mathcal{T}$ are retained for downstream use. The verification outcomes, including $\varphi$, $\gamma$, and examples of satisfied instances and violated instances, are subsequently incorporated into the next prompt to guide the LLM towards improved rule formalization.

**Example 1** *The following are examples of symbolic reasoning rules in the Game of 24 task.*

- $r_1 : \texttt{IsSmall}(x,y) \wedge \texttt{IsClose}(x,y) \rightarrow \texttt{Operation}(x,y,+)$. *This rule suggests applying addition to $x$ and $y$ if they are relatively small and numerically close.*

- $r_2 : \texttt{IsFactor}(z,24) \rightarrow \phi$. *It means that if the result $z$ is a factor of 24, it is allowed regardless of the action.*

- $r_3 : \texttt{IsFactor}(x,y,24) \rightarrow \texttt{Operation}(x,y,\times)$. *It suggests applying multiplication to $x$ and $y$ when they are the factors of 24.*

**Analysis:** Although we employ the LLMs for node labeling and rule formalization, our approach differs from the uncontrollable method that directly generates rules from reasoning trajectories using LLMs. By grounding rules in explicit matching criteria, we achieve greater consistency, interpretability, and reliability. Rule extraction is performed offline, and although frequent subgraph mining is theoretically NP-hard, the reasoning graphs in practice are sufficiently small to make the process tractable. Moreover, LLM calls during rule formalization are restricted to only a few subgraphs (typically fewer than ten), rendering the overall cost of rule mining negligible in comparison with online reinforcement learning.

## 4.2 REINFORCEMENT LEARNING FOR LLMS

### 4.2.1 RULE-BASED REWARD CONSTRUCTION

Given a set of verifiable symbolic reasoning rules $\mathcal{R} = \{r_1, r_2, \ldots, r_K\}$ from the reasoning trajectories, we integrate them as supervision signals into the reinforcement learning to guide policy optimization. Each rule $r_i \in \mathcal{R}$ is served as a reward function $r_\phi(y_s, r_i)$, which assigns scalar feedback to each reasoning step $y_s$ in a generated trajectory $o = \{y_1, y_2, \ldots, y_T\}$ based on whether $y_s$ satisfies the rule. Specifically, we define:

$$r_\phi(y_s, r_i) = \begin{cases} \rho, & \text{if } y_s \vDash r_i \\ 0, & \text{otherwise} \end{cases}, \tag{4}$$

where $\rho$ is a predefined reward value, which is set 1 in the positive rules extracted from successful trajectories, or $-1$ in the negative rules extracted from failed trajectories.

In order to compute the rule-based process reward, we aggregate scalar feedback from relevant rules. Rather than averaging over the entire rule set $\mathcal{R}$, we restrict computation to the subset of activated rules $\mathcal{R}_u = \{r_i : X \rightarrow A \mid y_s \vDash X\}$, where $y_s$ satisfies the precondition of each rule. This avoids diluting the supervision with unrelated rules, especially when $\mathcal{R}$ is large. Accordingly, we define the rule-based process reward as:

$$r_\phi(y_s, \mathcal{R}) = \frac{1}{|\mathcal{R}_u|} \sum_{i=1}^{K} \omega_i r_\phi(y_s, r_i), \tag{5}$$

where $K$ is the number of rules in $\mathcal{R}$ and $\omega_i$ is a rule weight.

**Example 2** *Consider a reasoning trajectory consisting in Game of 24 task with three steps:* $y_1$ : $1 + 5 = 6, left : \{6, 6, 10\}$, $y_2 : 10 - 6 = 4, left : \{4, 6\}$, *and* $y_3 : 6 \times 4 = 24, left : \{24\}$. *We analyze each reasoning step to identify the sets of rules it satisfies and activates, respectively, as illustrated in Example 1. Assuming $\rho = 1$ and $\omega_i = 1$ to each activated rule, the process reward is computed as follows: (1) for $y_1$: $y_1 \models r_1$, $y_1 \models r_2$; and $\mathcal{R}_u = \{r_1, r_2\}$. Thus $r_\phi(y_1, \mathcal{R}) = \frac{1}{2} \times (1 + 1) = 1$; (2) for $y_2$: $y_2 \models r_2$; and $\mathcal{R}_u = \{r_1, r_2\}$. Thus $r_\phi(y_2, \mathcal{R}) = \frac{1}{2} \times 1 = \frac{1}{2}$ ; and for $y_3$: $y_3 \models r_2$, $y_3 \models r_3$; and $\mathcal{R}_u = \{r_1, r_2, r_3\}$. Thus $r_\phi(y_3, \mathcal{R}) = \frac{1}{3} \times (1 + 1) = \frac{2}{3}$.*

**Verifiability**: Each symbolic reasoning rule has a well-defined precondition and consequence, and the matching relation $y_s \models r$ is binary and computable, which ensures safe and verifiable reward assignment. Since the rules are extracted directly from both successful and failed trajectories, the resulting reward signals are inherently grounded in empirical evidence, while support and confidence metrics further establish their reliability across trajectories. As a result, the rule-based reward function is fully computable, transparent, and auditable, which enables reproducible reward computation beyond the reach of opaque or purely learned models.

### 4.2.2 ADAPTIVE RULE WEIGHTING

However, the extracted rules exhibit obvious variations in both generality and predictive reliability. Broad rules (*e.g.*, $r_1$) are frequently activated across reasoning steps but offer weaker signals for task success, whereas specific rules (*e.g.*, $r_3$) occur less but provide more reliable indicators of correct reasoning. The uniform treatment of all rules fails to capture these distinctions, which results in suboptimal reward shaping and overoptimization. This issue is exacerbated during training, as the evolving policy alters the distribution of reasoning trajectories.

In order to address this limitation, we define a *reward informativeness metric* that quantifies the utility of each rule under the current policy. This metric enables dynamic adjustment of rule weights, thereby allowing the reward function to prioritize more informative rules during training. Specifically, the informativeness of rule $r_i$ at the $\tau$-th policy update iteration is defined as a weighted sum of its hit rate $\text{Hit}_\text{R}$ and success rate $\text{Succ}_\text{R}$:

$$\mathcal{I}^{(\tau)}(r_i) = \alpha \cdot \underbrace{\frac{1}{|\mathcal{T}^{(\tau)}|} \sum_{y_s \in \mathcal{T}^{(\tau)}} \mathbb{I}\{y_s \models r_i\}}_{\text{hit rate Hit}_\text{R}} + \beta \cdot \underbrace{\frac{\sum_{y_s \in \mathcal{T}^{(\tau)}} \mathbb{I}\{y_s \models r_i\} \cdot Succ(y_s)}{\sum_{y_s \in \mathcal{T}^{(\tau)}} \mathbb{I}\{y_s \models r_i\}}}_{\text{success rate Succ}_\text{R}} \tag{6}$$

where $|\mathcal{T}^{(\tau)}|$ denotes the set of reasoning steps sampled under the current policy, $\mathbb{I}\{y_s \models r_i\}$ is an indicator function that equals 1 if step $y_s$ satisfies rule $r_i$, and 0 otherwise, and $Succ(y_s) \in \{0, 1\}$ indicates whether step $y_s$ eventually leads to the correct answer.

Mathematically, the first term (hit rate Hit) penalizes overly specific rules that rarely trigger, preventing overfitting to sparse patterns, while the second term (success rate) penalizes broad rules that fail to distinguish between correct and incorrect reasoning paths. The hyperparameters $\alpha$ and $\beta$ control the trade-off between rule generality and reliability.

We then adaptively update the rule weight based on the informativeness gain between iterations: $\omega_i^{(\tau+1)} = \omega_i^{(\tau)} + \eta \cdot \Delta\mathcal{I}^{(\tau)}(r_i)$, where $\Delta\mathcal{I}^{(\tau)}(r_i) = \mathcal{I}^{(\tau)}(r_i) - \mathcal{I}^{(\tau-1)}(r_i)$ and $\eta$ is a learning rate. This update rule functions as a momentum-based adjustment: positive $\Delta\mathcal{I}^{(\tau)}$ implies that the rule is becoming more aligned with the current policy's successful trajectories, justifying a weight increase to reinforce this behavior. Conversely, a negative $\Delta\mathcal{I}^{(\tau)}$ signals that the rule is becoming either irrelevant or misleading as the policy shifts, prompting a reduction in its influence.

### 4.2.3 ADVANTAGE ESTIMATION AND POLICY UPDATE

After obtaining rule-based rewards, we incorporate the rule-based reward into the conventional outcome reward, yielding a rule-augmented outcome reward. Specifically, for each question, we sample a set of reasoning trajectories $\{o_1, o_2, \ldots, o_G\}$ from the old policy model $\pi_{\theta_{\text{old}}}$. The final reward $r_i$ for each trajectory $o_i$ consists of two components:

$$r_i = r_\phi(o_i, \mathcal{R}) + r_o(o_i) \tag{7}$$

where $r_o(o_i)$ is a scalar reward based on task outcomes. $r_\phi(o_i, \mathcal{R}) = \sum r_\phi(y_s, \mathcal{R})$ is a rule-based reward computed via explicit rules, ranging in [0,1], and can be efficiently obtained through a rule engine or lightweight validation function. We then apply within-batch normalization to the combined rewards $\mathbf{r} = \{r_1, \ldots, r_G\}$ Shao et al. (2024) :

$$\tilde{r}_i = \frac{r_i - \text{mean}(\mathbf{r})}{\text{std}(\mathbf{r})}. \tag{8}$$

Crucially, for advantage estimation in policy gradient updates (e.g., in PPO or REINFORCE-style objectives), we adopt a trajectory-level credit assignment strategy consistent with prior RLHF work Shao et al. (2024); Hu et al. (2025). Specifically, all tokens in trajectory $o_i$ share a uniform advantage estimate equal to the normalized composite reward:

$$\hat{A}_{i,t} = \tilde{r}_i, \quad \forall t \in \text{tokens}(o_i). \tag{9}$$

This unified formulation (Eq. 8–9) ensures that the rule-based signal is seamlessly integrated into the policy gradient without requiring a learned critic or value function, thereby preserving differentiability, avoiding bias from value approximation error, and maintaining compatibility with off-the-shelf RL pipelines Shao et al. (2024). The proof for unbiasedness of the advantage estimate in Appendix A.10.

## 5 EXPERIMENTS

Table 1: Performance comparison of different methods. RePAIR is our proposed rule-based method, while RePAIR$^-$ is a variant without adaptive rule weighting. "*" indicates results after SFT.

| Method | Game of 24 | Blocksworld | GSM8K | Avg. | $\Delta$ ($\uparrow$) |
|---|---|---|---|---|---|
| ***Qwen2.5-0.5B-Instruct*** | | | | | |
| Base | 33.00* | 24.00* | 25.26 | 27.42 | - |
| GRPO | 42.60 | 25.00 | 34.69 | 34.10 | +6.68 |
| GRPO w/ RePAIR$^-$ | 43.00 | 25.40 | **35.01** | 34.47 | +7.05 |
| GRPO w/ RePAIR | **45.00** | **26.00** | **35.01** | **35.34** | **+7.92** |
| Dr.GRPO | 47.00 | 25.00 | 33.82 | 35.27 | +7.85 |
| Dr.GRPO w/ RePAIR$^-$ | 47.00 | 26.20 | 34.02 | 35.74 | +8.32 |
| Dr.GRPO w/ RePAIR | **55.00** | **27.00** | **34.19** | **38.73** | **+11.31** |
| REINFORCE++ | 46.80 | 25.00 | 34.89 | 35.56 | +8.14 |
| REINFORCE++ w/ RePAIR$^-$ | 48.00 | 25.60 | 33.75 | 35.78 | +8.36 |
| REINFORCE++ w/ RePAIR | **51.00** | **26.00** | **35.48** | **37.49** | **+10.07** |
| ***Qwen2.5-Math-1.5B*** | | | | | |
| Base | 35.00* | 26.00* | 75.58 | 45.53 | - |
| GRPO | 50.40 | 29.00 | 75.43 | 51.61 | +6.08 |
| GRPO w/ RePAIR$^-$ | 52.80 | 30.00 | 76.04 | 52.95 | +7.42 |
| GRPO w/ RePAIR | **56.60** | **30.00** | **76.34** | **54.31** | **+8.78** |
| Dr.GRPO | 56.60 | 29.00 | 75.51 | 53.70 | +8.17 |
| Dr.GRPO w/ RePAIR$^-$ | 59.00 | 29.00 | 75.58 | 54.53 | +9.00 |
| Dr.GRPO w/ RePAIR | **64.20** | **30.00** | **75.73** | **56.64** | **+11.11** |
| REINFORCE++ | 57.60 | 29.00 | 75.89 | 54.16 | +8.63 |
| REINFORCE++ w/ RePAIR$^-$ | 58.80 | 29.40 | 76.04 | 54.75 | +9.22 |
| REINFORCE++ w/ RePAIR | **59.80** | **30.00** | **76.04** | **55.28** | **+9.75** |

### 5.1 EXPERIMENTAL SETUPS

In order to comprehensively evaluate the effectiveness of our proposed method, we selected language models of varying scales and several representative reinforcement learning algorithms as baselines.

**Foundational Models:** We apply it to two open-source models of different sizes to demonstrate the scalability and model-agnostic nature of our approach: (1) Qwen2.5-0.5B-Instruct Yang et al.

(2025), a lightweight, instruction-tuned model; (2) Qwen2.5-Math-1.5B Yang et al. (2024a), a model specifically optimized for the mathematical domain. In resource-intensive reinforcement learning, the smaller models reduce computational costs, enable faster iteration and large-scale experimentation, and provide a more controllable environment for validating reward modeling and rule-based.

**Reinforcement Learning Algorithms:** We benchmark our method against three reinforcement learning algorithms to ensure a fair and thorough comparison, including GRPO Shao et al. (2024), Dr.GRPO Liu et al. (2025), and REINFORCE++ Hu et al. (2025). Similarly to GRPO, we modify only the advantage estimation functions in each RL algorithm.

**Evaluation Benchmarks:** We assess model performance on five reasoning benchmarks, including mathematical games (Game of 24 Yao et al. (2023)), planning tasks (Blocksworld Valmeekam et al. (2023)), and diverse mathematical problem sets (GSM8K Cobbe et al. (2021), AIME24 AI-MO (2024a), AMC23 AI-MO (2024b)). We report the accuracy (%) on each benchmark.

**Implementation Details:** All experiments are conducted on a system equipped with 2 * NVIDIA A100 (40G) GPUs. Each trained model is evaluated 5 times and reports the average results. Further details on automatic rule extraction and reinforcement learning are provided in Appendix A.3.

## 5.2 MAIN RESULTS

We evaluated simple benchmarks on both 0.5B and 1.5B models, with the results summarized in Table 1. For more challenging mathematical benchmarks, due to the limitations of the smaller models, we conducted experiments only on the 1.5B model, and the corresponding results are presented in Table 2. There are several key trends that can be observed from the results as follows:

**RePAIR surpasses all competing methods across most evaluated tasks.** Specifically, RePAIR delivers substantial performance gains of 9.83% for the base models and 2.23% for other competitive RL algorithms without rules on average, highlighting its effectiveness and scalability. Even on the challenging AIME24 benchmark, RePAIR brings notable improvements, such as a 3.33 gain within the Dr.GRPO framework.

Table 2: Performance comparison of different methods across complex math reasoning benchmarks on Qwen2.5-Math-1.5B.

| Method | AIME24 | AMC23 | Avg. | $\Delta$ (↑) |
|---|---|---|---|---|
| Base | 10.20 | 56.71 | 33.46 | - |
| GRPO | 13.33 | 57.50 | 35.42 | +1.96 |
| GRPO w/ RePAIR | **13.33** | **58.75** | **36.04** | **+2.58** |
| Dr.GRPO | 11.11 | 57.08 | 34.10 | +0.64 |
| Dr.GRPO w/ RePAIR | **14.44** | **58.75** | **36.60** | **+3.14** |
| REINFORCE++ | 13.33 | 57.08 | 35.21 | +1.75 |
| REINFORCE++ w/ RePAIR | **14.44** | **58.54** | **36.49** | **+3.03** |

**Adaptive rule weighting is more effective than fixed weights.** As shown in Table 1, the RePAIR yields substantially greater performance gains compared to RePAIR⁻, a variant without adaptive rule weighting. This suggests that dynamically adjusting rule weight provides more effective reward shaping, leading to improved policy optimization.

**RePAIR is an algorithm-agnostic and universally applicable enhancement module.** RePAIR contributes consistently regardless of the policy update method and model size, which indicates that RePAIR is a general plug-in for almost any RL algorithm for any LLM.

**RePAIR generalizes across tasks without handcrafted rewards.** RePAIR demonstrates robust performance across diverse reasoning tasks without relying on task-specific reward engineering, as its rules are automatically extracted from model behaviors.

Table 3: Comparison of model performance on Game of 24 task using unverified rules (RULE) versus our curated rules (RePAIR) with Qwen2.5-Math-1.5B.

| Method | GRPO | Dr. GRPO | REINFORCE++ |
|---|---|---|---|
| RULE | 56.2 | 60.2 | 59.2 |
| **RePAIR (ours)** | **56.6** | **64.2** | **59.8** |
| $\Delta$ (↑) | +0.40 | +4.00 | +0.60 |

**RePAIR performs better in highly structured yet reward-sparse tasks.** In the combinatorial and reward-sparse Game of 24 task, RePAIR achieves the largest performance gain among competitive RL algorithms, with an improvement of 5.1%, demonstrating its effectiveness in guiding exploration and handling sparse rewards.

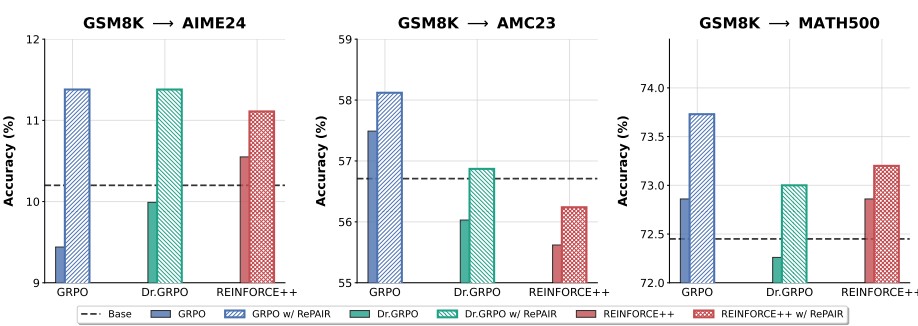

Figure 3: Out-of-distribution performance across different methods.

### 5.3 ANALYSIS

**Comparison of Different Rules:** To validate the quality of rules generated by RePAIR, we compare its performance against a baseline RULE that uses naively extracted, unfiltered rules. As shown in Table 3, the validated rules from RePAIR consistently outperform the unverified rules from RULE. Notably, when applied to Dr.GRPO, RePAIR achieves a substantial improvement of 4%. These results highlight that **our rule validation provides higher-quality training signals and leads to more effective policy optimization**. Meanwhile, we observe that increasing the number of rules does not necessarily lead to better learning performance. This suggests that reinforcement learning struggles to exploit all available rules, whereas a smaller subset of high-quality rules offers more stable and clearer learning signals.

**Model Generalization:** In order to assess the out-of-distribution (OOD) generalization capabilities of RePAIR, we train the Qwen2.5-Math-1.5B on the GSM8K and evaluate it on three unseen reasoning benchmarks: AIME24 Li et al. (2024), AMC23 Li et al. (2024), and Math500 Hendrycks et al. (2021). As illustrated in Figure 3, our method consistently outperforms all baseline approaches on these tasks, demonstrating that **RePAIR does not rely on overfitting and exhibits effective generalization beyond the training distribution**. More experiments are provided in Appendix A.4.

**Effects of RePAIR on Training Process:** We compare the test accuracy of different methods across different gradient steps to analyze the effects of rule-based process rewards on the training process. As shown in Figure 4, RePAIR leads to better performance as the training step increases, which indicates that **the model trained by RePAIR effectively learns to align with injected rules**.

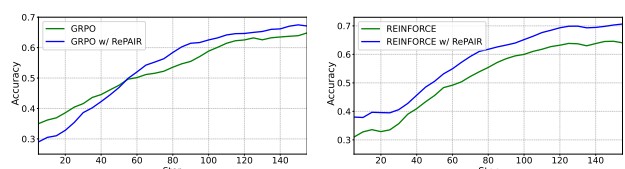

Figure 4: Comparison of performance of accuracy on the training process with Qwen2.5-Math-1.5B.

**Effects of RePAIR on Reasoning Behavior:** We evaluate the reasoning trajectories generated by LLMs trained with the baseline and RePAIR to assess the impact of symbolic rule supervision on model behavior. For each rule, we compute its support, confidence, and success rate on these reasoning trajectories. Table 5 shows the average results for all rules based on GRPO, which reveals that RePAIR

Table 4: Evaluation of reasoning trajectories under three rule-based metrics on the Game of 24 task with Qwen2.5-Math-1.5B.

| Method | Support | Confidence | $\text{Succ}_R$ |
|---|---|---|---|
| GRPO | 0.39 | 0.51 | 0.44 |
| **GRPO w/ RePAIR** | **0.40** | **0.53** | **0.54** |
| $\Delta$ ($\uparrow$) | +0.01 | +0.02 | +0.10 |

does not increase the number of rule activations, as the support remains similar. However, RePAIR substantially improves the $\text{Succ}_R$, indicating that **RePAIR teaches the model to apply rules more accurately and contextually instead of creating new rules**. This reveals that RePAIR improves the semantic alignment between symbolic rules and the model's decision-making process, leading to more reliable reasoning trajectories. More detailed experiments are provided in Appendix A.5.

**Efficiency of automatic rule extraction:** Table 5 reports the number of extracted rules and runtime across different benchmarks. As discussed in Section 4.1.2, the frequent subgraph mining operates on relatively small-scale data, resulting in a limited number of subgraphs (*i.e.,* candidate rules), while most of the runtime is consumed by LLM calls for rule formalization. After validation, the retained rules are compact yet high-quality, which reduces computational overhead and improves the effectiveness of reinforcement learning.

Table 5: Rule extraction statistics and runtime across benchmarks.

| Benchmark | Rules[#] | Time[sec] |
|---|---|---|
| **Game of 24** | 9 | 36.3 |
| **Blocksworld** | 5 | 25.2 |
| **GSM8K** | 4 | 12.2 |
| **AIME24** | 4 | 27.4 |
| **AMC23** | 4 | 14.6 |

## 6 CONCLUSION

We proposed RePAIR, a rule-based process-adaptive reinforcement learning framework, which automatically extracts symbolic reasoning rules from LLM-generated reasoning trajectories, enabling fine-grained and interpretable supervision. Extensive experiments across multiple tasks demonstrate that RePAIR yields significant improvements and serves as a general plug-in compatible with a wide range of RL algorithms and LLMs. By introducing symbolic rules, we enhance reinforcement learning for LLMs, making it more robust, interpretable, and scalable. This also opens new directions in automated reward design and symbolic process supervision for complex generative environments.

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

# A  APPENDIX

## A.1  REINFORCEMENT LEARNING FOR LLMS

Algorithm 1 outlines the Reinforcement Learning Stage, where the policy is iteratively optimized using both outcome and process-level rewards. At each iteration, responses are sampled from the current policy, and corresponding rewards are computed based on a predefined rule set R. Rule weights are adaptively updated according to their informativeness and impact on learning. The final policy is refined via a GRPO-based objective, enabling efficient reward shaping and stable policy improvement.

---

**Algorithm 1** Reinforcement Learning

**Input**: Large Language model $\pi_{\theta_{\text{init}}}$, outcome reward verifier $r_o$, rule set $\mathcal{R}$, sample number $K$, weight update rate $\eta$, total iteration $N$
**Output**: Optimized policy $\pi_\theta$
1: Initialize policy $\pi_\theta \leftarrow \pi_{\theta_{\text{init}}}$
2: Initialize rule weights $\omega_i = 1$ for all $r_i \in \mathcal{R}$
3: **for** each iteration $\tau = 1, 2, \ldots, N$ **do**
4:     Sample $K$ trajectories: $\{o^1, \ldots, o^K\} \sim \pi_\theta$
5:     Compute outcome rewards: $r_o(o^{1:K})$
6:     Compute process rewards: $r_\phi(y_s, \mathcal{R})$ with Eq. 5
7:     **for** each rule $r_i \in \mathcal{R}$ **do**
8:         Compute informativeness $\mathcal{I}^{(\tau)}(r_i)$ with Eq. 6
9:         Update rule weight: $\omega_i^{\tau+1} \leftarrow \omega_i^\tau + \eta \cdot \Delta\mathcal{I}^{(\tau)}(r_i)$
10:     **end for**
11:     Estimate advantage $A$ with Eq. 9
12:     Update policy: $\pi_\theta \leftarrow \arg\max_\theta \mathcal{J}_{\text{GRPO}}(\theta)$
13: **end for**
14: **return** optimized policy $\pi_\theta$

---

## A.2  EVALUATION BENCHMARK

- Game of 24 Yao et al. (2023): A numerical reasoning task that requires generating an arithmetic expression using four given numbers to reach 24.

- Blocksworld Valmeekam et al. (2023): An embodied planning benchmark where an agent must reach a specific block stacking arrangement from an initial state through moving operations such as PickUp and Stack.

- GSM8k Cobbe et al. (2021): A math word problem dataset that emphasizes multi-step numerical reasoning and arithmetic comprehension.

- AIME24 AI-MO (2024a): The AIME24 dataset is a collection of challenging problems from the 2024 American Invitational Mathematics Examination (AIME).

- AMC23 AI-MO (2024b): The AMC23 dataset is a benchmark derived from the American Mathematics Competitions, designed to evaluate and enhance the reasoning abilities of AI models on complex mathematical problems.

## A.3  IMPLEMENTATION DETAILS

For automatic rule extraction, we use Deepseek-R1 DeepSeek-AI et al. (2025) to generate 100 reasoning trajectories on each task and utilize GPT-4o Hurst et al. (2024) to formalize symbolic reasoning rules that have been validated to meet support $\varphi > 0.2$ and confidence $\gamma > 0.6$. We employ a task-specific training strategy. Due to the strict output format requirements of the Game of 24 and Blocksworld tasks, we first perform Supervised Fine-Tuning (SFT) on the models for these tasks before reinforcement learning. In contrast, for the GSM8K task, models are trained directly with reinforcement learning without preliminary SFT. In order to ensure a fair comparison across all methods, we maintain a consistent configuration for the RL training process. For each training

prompt, 8 responses (rollouts) are sampled. We use a batch size of 32 for all RL experiments. Hyperparameters are set as $\alpha = 0.5$, $\beta = 0.5$, and $\eta = 0.1$. All experiments are conducted on a system equipped with 2 * NVIDIA A100 (40G) GPUs. Each trained model is evaluated 5 times and reports the average results.

## A.4 ANALYSIS OF MODEL GENERALIZATION

In order to further evaluate the generalization ability of the model, we constructed a more challenging task, full Blocksworld, to assess the model's performance after training. By varying the minimum number of steps needed for a solution, we create a set of test cases with varying difficulty levels. As shown in Table 6, we observe that smaller models (*e.g.,* with 0.5B and 1.5B parameters) do not exhibit performance improvements with the injected symbolic reasoning rules; in some cases, their performance may even deteriorate. This result suggests that small models tend to overfit the rules present in the training data due to their limited capabilities. Instead of learning the underlying principles behind the rules, these models memorize them as rigid templates. Consequently, when deployed on out-of-distribution tasks, such templates not only fail to generalize but may even conflict with the correct problem-solving logic. In contrast, when we apply larger trained models (*e.g.,* with 3B parameters) on full Blocksworld, RePAIR performs better than the baseline. This indicates that as the model's capabilities improve, it can better learn the general principles introduced by the rules, thereby enabling more robust generalization to unseen problems. **The capabilities of RePAIR scale as the base model becomes more powerful.**

Table 6: Comparison of different models in full Blocksworld

| Model | Dr.GRPO | Dr.GRPO w/ RePAIR | $\Delta$ ($\uparrow$) |
|---|---|---|---|
| Qwen2.5-0.5B-Instruct | 27.50 | 26.15 | -1.35 |
| Qwen2.5-Math-1.5B | 38.40 | 38.40 | +0.00 |
| Qwen2.5-3B | 46.92 | 47.69 | +0.77 |

## A.5 EFFECTS OF REPAIR ON REASONING BEHAVIOR

In order to investigate the effects of RePAIR on reasoning behavior, we compare reasoning trajectories across models in the Game of 24 task with six rules used in the training process. The detailed results are shown in Table 7. Although RePAIR does not lead to a notable increase in the number of rule activations, it yields a substantial gain in success rate. This indicates that the model learns to selectively apply rules that are more effective, thereby prioritizing rule quality over mere frequency of usage.

Table 7: Evaluation of reasoning trajectories on the Game of 24 task with Qwen2.5-Math-1.5B.

| Method | Acc. | Support | | | | | | Confidence | | | | | | $\text{Succ}_R$ | | | | | |
|---|---|---|---|---|---|---|---|---|---|---|---|---|---|---|---|---|---|---|---|
| | | $r_1$ | $r_2$ | $r_3$ | $r_4$ | $r_5$ | Avg. | $r_1$ | $r_2$ | $r_3$ | $r_4$ | $r_5$ | Avg. | $r_1$ | $r_2$ | $r_3$ | $r_4$ | $r_5$ | Avg. |
| GRPO | 0.50 | 0.82 | 0.26 | 0.29 | 0.19 | 0.39 | 0.39 | 0.82 | 0.31 | 0.29 | 0.49 | 0.65 | 0.51 | 0.50 | 0.47 | 0.45 | 0.34 | 0.45 | 0.44 |
| GRPO w/ RePAIR | 0.59 | 0.81 | 0.25 | 0.32 | 0.21 | 0.43 | 0.40 | 0.81 | 0.28 | 0.32 | 0.54 | 0.70 | 0.53 | 0.59 | 0.54 | 0.55 | 0.48 | 0.53 | 0.54 |
| Dr.GRPO | 0.57 | 0.81 | 0.25 | 0.31 | 0.21 | 0.43 | 0.40 | 0.81 | 0.29 | 0.31 | 0.50 | 0.69 | 0.52 | 0.57 | 0.52 | 0.52 | 0.41 | 0.52 | 0.51 |
| Dr.GRPO w/ RePAIR | 0.67 | 0.80 | 0.27 | 0.32 | 0.23 | 0.41 | 0.41 | 0.80 | 0.30 | 0.32 | 0.52 | 0.71 | 0.53 | 0.67 | 0.62 | 0.59 | 0.48 | 0.61 | 0.60 |
| REINFORCE++ | 0.56 | 0.82 | 0.27 | 0.29 | 0.20 | 0.42 | 0.40 | 0.82 | 0.31 | 0.29 | 0.51 | 0.68 | 0.52 | 0.56 | 0.51 | 0.49 | 0.39 | 0.51 | 0.49 |
| REINFORCE++ w/ RePAIR | 0.61 | 0.83 | 0.27 | 0.32 | 0.21 | 0.42 | 0.41 | 0.83 | 0.32 | 0.32 | 0.50 | 0.70 | 0.53 | 0.62 | 0.57 | 0.58 | 0.45 | 0.56 | 0.56 |

## A.6 ADDITIONAL EXPERIMENTAL RESULTS WITH 3B MODELS

To demonstrate the scalability of our approach, we extended our evaluation to the larger Qwen2.5-3B-Instruct model. As shown in Table 8, RePAIR consistently enhances performance across all baseline methods. Notably, when integrated with GRPO, it achieves a significant improvement of 2.58%. These results, complementing our findings on smaller models, confirm that our method is effective and scalable across models of varying sizes.

Table 8: Performance comparison of different methods on Qwen2.5-3B-Instruct.

| Method | AMC23 | Math500 | Avg. | $\Delta$ ($\uparrow$) |
|---|---|---|---|---|
| Base | 41.67 | 62.07 | 51.87 | - |
| GRPO | 42.29 | 63.67 | 52.98 | +1.11 |
| GRPO w/ RePAIR | **44.17** | **64.73** | **54.45** | **+2.58** |
| Dr.GRPO | 41.88 | 63.47 | 52.68 | +0.81 |
| Dr.GRPO w/ RePAIR | **42.92** | **64.67** | **53.80** | **+1.93** |
| REINFORCE++ | 42.50 | 63.33 | 52.92 | +1.05 |
| REINFORCE++ w/ RePAIR | **43.54** | **63.80** | **53.67** | **+1.80** |

## A.7 COMPARISON WITH LLM-BASED PRMS

Table 9: Performance comparison between RePAIR and LLM-based Process Reward Models (PRMs). * denotes the results from Cui et al. (2025).

| Method | Reward Model Size | AIME 24 | AMC 23 | Avg. |
|---|---|---|---|---|
| | Qwen2.5-3B | 10.70 | 44.00 | 27.35 |
| PRIME* | Qwen2.5-7B | 13.20 | 42.90 | 28.05 |
| | Qwen2.5-14B | 10.80 | 44.10 | 27.45 |
| **RePAIR (Ours)** | – | **13.33** | **44.84** | **29.08** |

Recent studies have increasingly adopted Large Language Models (LLMs) as Process Reward Models (PRMs) to guide reasoning steps. To evaluate our approach against this paradigm, we conducted a comparative experiment with PRIME(Cui et al., 2025), a method that employs LLMs of varying sizes (Qwen2.5-3B, 7B, and 14B) as reward models.

The results, summarized in Table 1, demonstrate that RePAIR outperforms these PRM-based methods across all metrics. Notably, RePAIR achieves a higher average score (29.08) than PRIME even when the latter utilizes a 14B parameter reward model. Crucially, unlike these approaches that require maintaining and querying a separate, often computation-heavy LLM to serve as a reward model, RePAIR operates without an auxiliary model during training. Consequently, our method consumes significantly less computational power while achieving superior performance.

## A.8 EXPERIMENTS ON LARGER MODELS

Table 10: Performance comparison of different methods on Qwen2.5-7B-Base.

| Method | AIME24 | AMC23 | Math500 | Avg. | $\Delta$ ($\uparrow$) |
|---|---|---|---|---|---|
| GRPO | 10.00 | 34.37 | 52.20 | 32.19 | - |
| GRPO w/ RePAIR | **13.33** | **44.84** | **57.80** | **38.55** | **+6.36** |
| Dr.GRPO | 3.33 | 35.62 | 50.56 | 29.83 | - |
| Dr.GRPO w/ RePAIR | **6.66** | **42.50** | **56.00** | **35.05** | **+5.22** |
| REINFORCE++ | 6.66 | 35.24 | 51.96 | 31.28 | - |
| REINFORCE++ w/ RePAIR | **13.33** | **41.25** | **55.76** | **36.78** | **+5.50** |

To investigate whether the efficacy of our proposed method extends to larger-scale architectures, we conducted additional experiments using the Qwen2.5-7B-Base model. This analysis aims to verify if the performance gains observed in smaller models are consistent as model capacity increases.

The results, presented in Table 10, demonstrate that our method maintains its effectiveness on the 7B parameter scale. As shown, integrating RePAIR consistently yields significant performance improvements across all evaluated baselines. Most notably, when combined with GRPO, RePAIR achieves an average improvement of 6.36 points across the AIME24, AMC23, and MATH500

benchmarks. Similarly, substantial gains of 5.22 and 5.50 points are observed with Dr.GRPO and REINFORCE++, respectively. These findings confirm that the benefits of our approach are robust and scale effectively to larger language models.

## A.9 SENSITIVITY ANALYSIS OF HYPERPARAMETERS $\alpha$, $\beta$ AND $\eta$

We conduct a comprehensive sensitivity analysis to examine the impact of the hyperparameters $\alpha$, $\beta$ and $\eta$ in our proposed informativeness metric (Eq. 6 ). The parameter $\alpha$ and $\beta$ balance the contribution of the hit rate and the success rate, while $\eta$ controls the learning rate for the subsequent rule weight updates.

We evaluated our model across a comprehensive grid of values for $\alpha \in \{0.3, 0.5, 0.7\}, \eta \in \{0.05, 0.10, 0.15\}$ on the game24, with all other experimental settings fixed. The results are summarized in the table 11:

Table 11: Model performance (accuracy) for different $(\alpha, \beta, \eta)$ pairs.

| $\alpha$ | 0.3 | 0.3 | 0.3 | 0.5 | 0.5 | 0.5 | 0.7 | 0.7 | 0.7 |
|---|---|---|---|---|---|---|---|---|---|
| $\beta$ | 0.7 | 0.7 | 0.7 | 0.5 | 0.5 | 0.5 | 0.3 | 0.3 | 0.3 |
| $\eta$ | 0.05 | 0.10 | 0.15 | 0.05 | 0.10 | 0.15 | 0.05 | 0.10 | 0.15 |
| **Accuracy** | 0.536 | 0.526 | 0.516 | 0.500 | **0.550** | 0.542 | 0.502 | 0.504 | 0.510 |

Our analysis reveals two key findings:

(1) **Robustness Across a Broad Range:** The model performance is relatively stable across most parameter combinations, with mean accuracy consistently above 0.50. This indicates that our method is not critically dependent on a finely-tuned $(\alpha, \eta)$ pair, which enhances its reproducibility and practical utility.

(2) **An Optimal Balance at $\alpha = 0.5$:** The best and most stable performance is achieved when $\alpha = 0.5$, paired with $\eta = 0.1$. We posit that this value strikes an effective balance in the informativeness metric. A lower $\alpha$ may overemphasize the *hit rate*, leading to frequent updates for rules that are triggered often but not necessarily correlated with success. Conversely, a higher $\alpha$ may overfit to the immediate *success rate* of the current policy, potentially stifling the exploration of diverse reasoning paths that could be beneficial in the long term. Therefore, $\alpha = 0.5$ provides an effective trade-off between encouraging rule diversity and leveraging successful trajectories.

## A.10 UNBIASEDNESS OF THE ADVANTAGE ESTIMATE

To verify the mathematical soundness and unbiasedness of the proposed advantage term $\hat{A}_{i,t} = \tilde{r}_i$ (Eq. 9), we proceed as follows, leveraging insights from Dr. GRPO Liu et al. (2025) and classical policy gradient theory.

### A.10.1 DEFINITION OF UNBIASED ADVANTAGE ESTIMATION

An advantage term $\hat{A}_{i,t}$ is unbiased for policy gradient updates if, under the current policy $\pi_\theta$, the conditional expectation of $\hat{A}_{i,t}$ given the state $s_t = (q, o_{i,<<t})$ is zero:

$$\mathbb{E}_{o_i \sim \pi_\theta(\cdot|q)} \left[ \hat{A}_{i,t} \mid s_t \right] = 0. \tag{10}$$

This condition ensures that the policy gradient estimates only capture the relative quality of trajectories (not systematic biases) and converges to the true policy gradient.

### A.10.2 PROOF OF UNBIASEDNESS

Unbiasedness of the Composite Reward The composite reward $r_i = r_\phi(o_i, \mathcal{R}) + r_o(o_i)$ integrates rule-based intrinsic rewards and outcome-based extrinsic rewards. Both components are computed via explicit, parameter-free functions:

- $r_o(o_i)$ is a deterministic function of the trajectory's final outcome (e.g., correct/incorrect for math problems), thus unbiased.
- $r_\phi(o_i, \mathcal{R}) = \sum r_\phi(y_s, \mathcal{R})$ is computed via a rule engine, directly quantifying intermediate reasoning validity without approximation.

Let $V(q) = \mathbb{E}_{o_i \sim \pi_\theta(\cdot|q)}[r_i]$ denote the true state value of question $q$ under policy $\pi_\theta$. Then:

$$\mathbb{E}_{o_i \sim \pi_\theta(\cdot|q)}[r_i] = V(q), \tag{11}$$

confirming $r_i$ is an unbiased estimate of $V(q)$.

**Unbiasedness of Normalized Reward** The within-batch normalization (Eq. 8) computes $\tilde{r}_i = \frac{r_i - \text{mean}(\mathbf{r})}{\text{std}(\mathbf{r})}$, where $\mathbf{r} = \{r_1, ..., r_G\}$ is the set of composite rewards for the batch. Since $\text{mean}(\mathbf{r}) = \frac{1}{G} \sum_{j=1}^{G} r_j$ is the sample mean of $r_i$, it holds that:

$$\mathbb{E}_{o_i \sim \pi_\theta(\cdot|q)}[\text{mean}(\mathbf{r})] = V(q), \tag{12}$$

(by linearity of expectation and Eq. 11). Substituting into the normalized reward:

$$\mathbb{E}_{o_i \sim \pi_\theta(\cdot|q)}[\tilde{r}_i] = \frac{\mathbb{E}[r_i] - \mathbb{E}[\text{mean}(\mathbf{r})]}{\text{std}(\mathbf{r})} = \frac{V(q) - V(q)}{\text{std}(\mathbf{r})} = 0. \tag{13}$$

**Unbiasedness of Trajectory-Level Advantage** By Eq. 9, all tokens in trajectory $o_i$ share the same advantage $\hat{A}_{i,t} = \tilde{r}_i$. Since $\tilde{r}_i$ is computed based on batch-level statistics (independent of the state $s_t = (q, o_{i,<<t})$), the conditional expectation satisfies:

$$\mathbb{E}_{o_i \sim \pi_\theta(\cdot|q)}[\hat{A}_{i,t} \mid s_t] = \mathbb{E}[\tilde{r}_i \mid s_t] = \mathbb{E}[\tilde{r}_i] = 0. \tag{14}$$

The proposed advantage term $\hat{A}_{i,t} = \tilde{r}_i$ satisfies the unbiasedness condition (Eq. 10) for policy gradient updates.

## A.11 OPTIMAL POLICY INVARIANCE

We show that, under the following assumptions, augmenting the outcome reward with the process reward does not change the set of optimal policies with respect to the ground-truth outcome reward.

- Assumption 1: the task outcome is binary and the outcome reward for output $o_i$ is $r_o(o_i) = Succ(o_i) \in \{0, 1\}$. The performance measure of interest is the accuracy rate of model inference under the current policy $\pi$, i.e., $\mu(\pi) = \mathbb{E}[Succ(o)]$;
- Assumption 2.: at the converged weights $\omega^*$, there exist constants $\mu_1, \mu_0 \in \mathbb{R}$ such that

$$\mu_1 \triangleq \mathbb{E}\big[r_\phi(o; \omega^*) \mid Succ(o) = 1\big], \qquad \mu_0 \triangleq \mathbb{E}\big[r_\phi(o; \omega^*) \mid Succ(o) = 0\big],$$

with $\mu_1 \geq \mu_0$, and for any policy $\pi$,

$$\mathbb{E}_{o \sim \pi}\big[r_\phi(o; \omega^*)\big] = \mu_1 \Pr_{o \sim \pi}(Succ(o) = 1) + \mu_0 \Pr_{o \sim \pi}(Succ(o) = 0).$$

In RePAIR, the adaptive weighting scheme leverages both the hit rate and the success rate to assign larger weights to rules that occur more frequently on successful trajectories. Consequently, once the rule weights have converged, successful trajectories receive at least as much process reward as unsuccessful ones.

**Theorem 1** *Given the shaped reward $r = r_o + r_\phi(\cdot; \omega^*)$, and assuming that assumptions 1 and 2 hold. Define the shaped-reward learning objective*

$$J_r(\pi) \triangleq \mathbb{E}_{o \sim \pi}\big[r_o(o) + r_\phi(o; \omega^*)\big],$$

*and recall the outcome-based learning objective and optimal policy set*

$$J_o(\pi) \triangleq \mathbb{E}_{o \sim \pi}\big[r_o(o)\big], \qquad \Pi^* \triangleq \arg\max_\pi J_o(\pi)$$

*Then the shaped-reward objective preserves the optimal policy set, i.e.,*

$$\arg\max_\pi J_r(\pi) = \arg\max_\pi J_o(\pi) = \Pi^*.$$

**Proof 1** *By definition, for any policy $\pi$,*

$$\mu(\pi) = \mathbb{E}[Succ(o)] = J_o(\pi).$$

*With assumption 2, the expected process reward under $\pi$ satisfies*

$$\mathbb{E}_{o\sim\pi}\big[r_\phi(o;\omega^*)\big] = \mu_1 \Pr(Succ(o) = 1) + \mu_0 \Pr(Succ(o) = 0) = \mu_1\mu(\pi) + \mu_0\big(1 - \mu(\pi)\big).$$

*Substituting this into the shaped objective $J_r(\pi)$, we obtain*

$$\begin{aligned}
J_r(\pi) &= \mu(\pi) + \mathbb{E}\big[G_p(\tau;\omega^*)\big] \\
&= \mu(\pi) + \big[\mu_1\mu(\pi) + \mu_0\big(1 - \mu(\pi)\big)\big] \\
&= \mu(\pi) + \mu_0 + (\mu_1 - \mu_0)\mu(\pi) \\
&= \mu_0 + \big[1 + (\mu_1 - \mu_0)\big]\mu(\pi).
\end{aligned}$$

*The term $\mu_0$ is a constant independent of $\pi$. Since $\mu_1 \geq \mu_0$, the coefficient*

$$c \triangleq 1 + (\mu_1 - \mu_0)$$

*satisfies $c \geq 1 > 0$. Therefore $J_r(\pi)$ is a strictly increasing affine function of $\mu(\pi)$:*

$$J_r(\pi) = const + c\,\mu(\pi), \qquad c > 0.$$

*Hence, for any two policies $\pi, \pi'$,*

$$\mu(\pi) > \mu(\pi') \quad \Longleftrightarrow \quad J_r(\pi) > J_r(\pi').$$

*Maximizing $J_r(\pi)$ over $\pi$ is therefore equivalent to maximizing $\mu(\pi)$ over $\pi$, and we obtain*

$$\arg\max_\pi J_r(\pi) = \arg\max_\pi \mu(\pi) = \Pi^*.$$

*Thus, the shaped reward $r = r_o + r_\phi(\cdot;\omega^*)$ preserves the outcome-optimal policy set.*

## A.12 DISCUSSION ON METHOD GENERALIZATION

The RePAIR framework is not inherently constrained to tasks with binary final outcomes. The core mechanism of our adaptive weighting relies on correlating rule application with a positive quality signal, which can be derived from various intermediate feedback sources beyond final ground-truth labels. We generalize the notion of "success rate" to suit broader domains:

**Code Generation**: "Success" can be defined via execution feedback (e.g., successful compilation, passing unit tests, or no runtime errors). Rules leading to executable code are upweighted, serving as a proxy for functional correctness.

**Formal Reasoning**: In theorem proving (e.g., Lean, Coq), we can utilize intermediate validity checks. Rules generating logically valid transitions or state changes accepted by the solver receive positive reinforcement.

**Open-Ended Dialogue**: For alignment tasks, feedback can stem from preference models or safety filters. Rules consistently producing safe or high-reward responses (as measured by an external Reward Model) are prioritized.

In essence, our method requires only a verifiable environmental signal—whether terminal or intermediate—to guide rule adaptation, making it flexible for diverse reasoning and generation tasks.

## A.13 REPRODUCIBILITY STATEMENT

We are committed to ensuring the reproducibility of our results. Accordingly, we provide the following information:

(1) Code Availability: All code for training, evaluation, and analysis is publicly available at: [https://anonymous.4open.science/r/RePAIR-8EFC]. The repository includes detailed README instructions for installation, configuration, and usage.

(2) Datasets: All datasets used in this paper are publicly available. We provide links and preprocessing scripts in the repository. No private or restricted-access data were used.

(3) Experimental Settings: The exact hyperparameters used in our experiments are listed in Appendix A.3. Random seeds for training and evaluation are explicitly specified, and multiple runs are reported to account for variance.

(4) Computational Resources: Experiments were conducted on 2 * NVIDIA A100 (40G) GPUs.

(5) Environment: The software environment (Python version, PyTorch/TensorFlow version, CUDA version) is specified in the repository. A requirements.txt file is included for easy environment setup.

