# OpenReview forum: "RePAIR: A Rule-based Process-Adaptive Reinforcement for Large Language Model Training"
_ICLR.cc/2026/Conference — ICLR 2026 Conference Withdrawn Submission_

### Official Review · Reviewer_cA2d · 2025-10-30

**Soundness:** 2
**Presentation:** 3
**Contribution:** 2
**Rating:** 6
**Confidence:** 3

**Summary:**

This paper introduces RePAIR, a Rule-based Process-Adaptive Reinforcement framework aimed at improving reinforcement learning for LLMs. It tackles the challenges of sparse and ambiguous reward signals in traditional RLHF and RLVR by incorporating symbolic reasoning rules that yield verifiable and adaptive process-level rewards. These rules are automatically extracted from model-generated reasoning trajectories via frequent subgraph mining and semantic summarization, enabling interpretable and fine-grained feedback throughout training.

**Strengths:**

1. The introduction of symbolic, rule-based process rewards enhances interpretability, verifiability, and adaptability, effectively addressing the limitations of outcome-based and black-box reward models.

2. The framework automatically derives reasoning rules from LLM trajectories, reducing dependence on manual reward engineering and human annotation.

3. RePAIR achieves consistent and substantial improvements across multiple reinforcement learning algorithms and reasoning benchmarks.

4. The paper is well written, with clear algorithmic descriptions, implementation details, and a stated plan for open-source release.

**Weaknesses:**

1. In Table 3, the performance gain over the unverified RULE baseline is modest for GRPO and REINFORCE++, suggesting that improvements may primarily stem from adaptive weighting rather than rule validation.

2. Experiments are conducted mainly on small-scale models (≤3B parameters), leaving it unclear whether RePAIR scales effectively to larger models (e.g., 7B+).

3. The influence of the informativeness update parameters ($\alpha$, $\beta$, $\eta$) is not systematically analyzed, which may limit reproducibility.

**Questions:**

1–3. See weaknesses above.

4. How does RePAIR generalize to tasks beyond mathematical or structured reasoning, such as commonsense or dialogue-based reasoning?

5. What is the computational overhead of rule extraction and verification relative to standard RL training?

6. The adaptive weighting mechanism depends on the success rate, which requires access to the final outcome. Does this constrain applicability in settings with only process-level feedback (i.e., without a final answer)?

7. In tasks where intermediate reasoning cannot be easily decomposed into discrete steps, how would graph construction and rule matching be adapted?

---

> ### Author Response · Authors · 2025-11-27
> **Response by authors for your concerns**
>
> Thank you for your thoughtful review! Here we address your questions about the work, and we welcome any further comments or questions.
> ## Q1:In Table 3, the performance gain over the unverified RULE baseline is modest for GRPO and REINFORCE++, suggesting that improvements may primarily stem from adaptive weighting rather than rule validation.
>
> Thanks for the comment! As pointed out, the performance gain over the unverified rule baseline seems modest. The main reason is that the unverified rule set contains a mixture of both high-quality rules that we need and low-quality (noisy) rules. Although the adaptive weighting mechanism can mitigate the negative impact of low-quality rules to some extent, it is not a complete solution.
>
> Without the rule validation step, the inclusion of excessively noisy rules significantly increases the computational overhead for reward calculation. Worse yet, the presence of unverified rules may lead to a substantial degradation in performance as observed in Table 3, where the score for **Dr.GRPO drops by 4 points** when using unverified rules. This evidence strongly suggests that rule validation remains a necessary component to ensure stability and efficiency, which prevents performance regression caused by noisy signals.
>
>  ## Q2:Experiments are conducted mainly on small-scale models (≤3B parameters), leaving it unclear whether RePAIR scales effectively to larger models (e.g., 7B+).
>
> Thanks for the comment! Due to computational resource constraints, our initial experiments focused on smaller-scale models.
>
> In order to verify whether our method scales effectively to larger models, we have added additional experiments on Qwen2.5-7B. The experimental settings remained consistent with those described in the paper. We evaluated the method on the AIME 24, AMC 23, and Math500 datasets, comparing it against the GRPO, Dr.GRPO, and REINFORCE++ baselines.
> The results are presented in the table below:
> | Method                | AIME24 | AMC23 | MATH500 | Avg.          |
> |-----------------------|--------|-------|---------|---------------|
> | GRPO                  | 10.00  | 34.37 | 52.20   | 32.19         |
> | GRPO w/ RePAIR        | **13.33**  | **44.84** | **57.80**   | **38.55** (+6.36)        |
> | Dr.GRPO               | 3.33   | 35.62 | 50.56   | 29.83         |
> | Dr.GRPO w/ RePAIR     | **6.66**   | **42.50** | **56.00**   | **35.05** (+5.22) |
> | REINFORCE++           | 6.66   | 35.24 | 51.96   | 31.28         |
> | REINFORCE++ w/ RePAIR | **13.33**  | **41.25** | **55.76**   | **36.78** (+5.50) |
>
> As can be seen from the table, our method consistently achieves significant performance improvements on the larger-scale model across all algorithms. Notably, RePAIR yielded an average increase of **6.36** points when combined with GRPO, which confirms that the **effectiveness of our approach scales well to larger models**.
>
>  ## Q3:The influence of the informativeness update parameters ($\alpha$, $\beta$, $\eta$) is not systematically analyzed, which may limit reproducibility.
>
> Thanks for the suggestion! We have added a comprehensive sensitivity analysis to examine their impact.
> As detailed in our paper (Section 4.2.2), $\alpha$ balances the contributions of the hit rate ($Hit_R$) and the success rate ($Succ_R$), while $\eta$ is the learning rate for the rule-weight update based on the informativeness gain.
> We evaluated our model across a comprehensive grid of values for $\alpha \in \{0.3, 0.5, 0.7\}, \eta \in \{0.05, 0.10, 0.15\}$ on the Game of 24 task, with all other experimental settings fixed. The results are summarized in the table below:
>
> | α| 0.3 | 0.3 | 0.3 | 0.5 | 0.5 | 0.5 | 0.7 | 0.7 | 0.7 |
> |-----|-----|-----|-----|-----|-----|-----|-----|-----|-----|
> | β| 0.7 | 0.7 | 0.7 | 0.5 | 0.5 | 0.5 | 0.3 | 0.3 | 0.3 |
> | η| 0.05| 0.10| 0.15| 0.05| 0.10| 0.15| 0.05| 0.10| 0.15|
> | **Accuracy** | 0.536| 0.526 | 0.516 | 0.500 | **0.550**| 0.542 | 0.502 | 0.504 | 0.510 |
>
> The key findings from this analysis are:
>
> **Robustness across a Broad Range**: The performance is relatively stable across most parameter combinations, with mean accuracy consistently above 0.50. This indicates that our method does not critically depend on a finely tuned ($\alpha$, $\eta$) pair and is therefore highly reproducible.
>
> **An Optimal Balance at $\alpha=0.5$** : The best and most stable performance is achieved when $\alpha=0.5$, paired with $\eta=0.1$. We hypothesize that this value strikes an effective balance: a lower $\alpha$ may overemphasize the hit rate, updating weights for rules that are frequently triggered but not necessarily helpful; a higher $\alpha$ may overfit to the success rate of the current policy, potentially reducing exploration. The value $\alpha=0.5$ optimally trades off between encouraging rule diversity and leveraging successful trajectories.
>
> Overall, these results confirm that our method is robust to the choice of ($\alpha$, $\eta$), alleviating the concern regarding reproducibility.

---

> > ### Author Response · Authors · 2025-11-27
> >
> > ## Q4:How does RePAIR generalize to tasks beyond mathematical or structured reasoning, such as commonsense or dialogue-based reasoning?
> >
> > Thanks for the question! Although the paper primarily focuses on structured reasoning tasks such as mathematics and logic that are central challenges in current LLM reasoning research[1][2], the RePAIR framework is generalizable to other domains. The main reason for this generalizability is that the principle of our method, extracting interpretable patterns from successful trajectories, applies beyond strict symbolic logic.
> > In order to demonstrate this scalability, we have added additional experiments on StrategyQA[2], a task requiring multi-hop reasoning and commonsense knowledge, which is more open-ended than pure math problems. As shown in the table below, RePAIR(ours) achieves a significant performance improvement over both the base model and the REINFORCE++ baseline.
> > | Method              | StrategyQA  |
> > |---------------------|-----------------------|
> > | Qwen2.5-3B-Instruct | 60.98                 |
> > | REINFORCE++         | 61.13                 |
> > | REINFORCE++ w/RePAIR              | **63.51**                 |
> >
> > These results demonstrate that our method **remains effective in broader reasoning scenarios such as commonsense reasoning.**
> >
> > ### References:
> > [1]Li, Zhong-Zhi et al. “From System 1 to System 2: A Survey of Reasoning Large Language Models.” ArXiv abs/2502.17419 (2025).
> >
> > [2]Cheng F, Li H, Liu F, et al. Empowering llms with logical reasoning: A comprehensive survey.IJCAI(2025)
> >
> > [3] Geva, Mor, et al. "Did aristotle use a laptop? a question answering benchmark with implicit reasoning strategies." Transactions of the Association for Computational Linguistics 9 (2021): 346-361.
> >
> >  ## Q5:What is the computational overhead of rule extraction and verification relative to standard RL training?
> >
> > Thanks for the question! As shown in Table 5, the time overhead for the rule extraction phase is minimal, typically taking only **tens of seconds**. This is negligible compared to the standard RL training process, which usually spans several hours. Furthermore, since the rule extraction module does not involve gradient updates or model training, it consumes very few computational resources (and can even run without a local GPU), which makes the overall additional overhead of RePAIR extremely low relative to the baseline RL training.
> >
> > ## Q6:The adaptive weighting mechanism depends on the success rate, which requires access to the final outcome. Does this constrain applicability in settings with only process-level feedback (i.e., without a final answer)?
> >
> > Thanks for the question! Although our implementation indeed utilizes the final outcome to calculate the "success rate" for weight adaptation, the architecture of RePAIR is not inherently constrained to binary final outcomes. The core mechanism relies on a quality signal to update rule weights, and this signal can be derived from various forms of intermediate process feedback when a final answer is unavailable:
> >
> > **Redefining "Success" as "Progress"**: In the absence of a final label, the "success rate" in our adaptive formula can be generalized to a "compliance rate" or "progress score." Any verifiable signal provided by the environment can serve as the ground truth for this update.
> >
> > **Domain-Specific Process Signals**:
> >
> > Coding Tasks: We can utilize execution feedback (e.g., successful compilation, passing unit tests, or lack of runtime errors) as a proxy for success. A rule that consistently leads to code that compiles is upweighted, even if the final functional correctness is unknown.
> >
> > Logical Reasoning: We can employ consistency checkers or formal verifiers (e.g., Lean or Coq provers) to validate intermediate steps. Rules that generate logically valid transitions receive positive reinforcement.
> >
> > Dialogue/Open-Ended Generation: We can integrate preference models or safety filters as the feedback mechanism. Rules that produce safe, coherent, or user-aligned responses (as measured by a reward model or heuristic) gain higher weights.
> >
> > Therefore, **our adaptive weighting mechanism is highly flexible**. It requires only a correlation between the rule's application and a positive environmental signal, whether that signal comes from a final answer or an intermediate process check. We will clarify this generalization capability in the appendix of the revised paper.

---

> > > ### Author Response · Authors · 2025-11-27
> > >
> > > ## Q7:In tasks where intermediate reasoning cannot be easily decomposed into discrete steps, how would graph construction and rule matching be adapted?
> > > Thanks for the question! For such tasks, we would first induce a discrete semantic graph over the raw reasoning trajectory and then apply the same rule mining on this induced graph. Specifically, given the model’s raw generation, an auxiliary LLM is prompted to segment the trace into a sequence of semantic units (e.g., clauses, tool calls, or high-level actions) and to assign each unit a coarse action label, for example, introduce assumption, invoke tool, update state, or draw conclusion. These units are then treated as graph nodes with their labels as attributes, and we connect them with temporal edges that reflect the order in the reasoning trajectories. Rule mining is applied to this induced graph in the same way as in the step-based setting. A systematic exploration of such adaptations is an interesting direction for future work.

---

### Official Review · Reviewer_4qNx · 2025-11-01

**Soundness:** 2
**Presentation:** 2
**Contribution:** 2
**Rating:** 4
**Confidence:** 4

**Summary:**

In this paper, the authors proposed an innovative framework called RePAIR. This framework can automatically extract symbolic rules and transform them into dynamic, fine-grained process reward signals. The authors compare the method with a series of baselines and evaluate its effectiveness.

**Strengths:**

1. In this paper, the authors provide sufficient experimental details and hyperparameters. These settings cover the settings for reproducibility.

2. The structure and format of this paper are clear and easy to follow. The authors also provide clear figures and tables to enhance the readability.

**Weaknesses:**

1. In the related work section, the authors mention PRM but fail to discuss these methods. In recent research, there are many methods of LLM-as-a-judge using strong LLMs or token-level process reward. These methods are direct competing solutions for the method. The lack of comparison between these methods weakens the persuasiveness of the method.

2. The experiments in this paper are primarily based on the Qwen model. Although they cover different scales of LLM, the model architecture is singular. This setting limits the generalizability of the method and contradicts the model-agnostic assumption mentioned in the paper. In addition, the method relies on external strong teacher models like GPT-4o or Deepseek-R1. This raises the question of whether the method remains effective when using weaker models for rule extraction.

**Questions:**

1. The authors mention that rules can be extracted from successful and failed trajectories. However, the authors only discuss positive rules. An issue worth discussing and studying is how negative rules have an impact. The authors may need to present a case study of negative rules.

2. In this paper, the authors use first-order logic to represent rules. This strategy is clear and verifiable. However, this hard-coded approach may limit the LLM's reasoning ability since the process of LLM reasoning may be more ambiguous. An open question is whether more ambiguous soft rules would lead to better results.

---

> ### Author Response · Authors · 2025-11-27
> **Response by authors for your concerns**
>
> Thank you for your thoughtful review and constructive feedback. Below, we address your concerns and questions in detail.
> ## Q1:The lack of comparison between other PRM methods weakens the persuasiveness of the method.
>
> Thanks for the comment! We acknowledge that many recent works utilize LLMs as judges or employ token-level Process Reward Models (PRMs). However, our approach differs fundamentally in its design philosophy and resource requirements. While standard PRMs often function as "black boxes" or require training dedicated models, RePAIR focuses on rule mining to explicitly guide the design of process rewards. The interpretable rules mined by our method can, in fact, serve as criteria for other PRM-based judgments, suggesting a complementary rather than purely competitive relationship.
>
> Furthermore, our method offers a significant advantage in data and computational efficiency. Unlike typical PRM approaches (e.g., [1]) that require training on hundreds of thousands of data points, RePAIR achieves substantial performance gains by mining rules from only a few hundred reasoning trajectories. This makes our approach feasible with a fraction of the data and computing resources required by standard PRMs.
>
> In order to address the lack of direct comparison mentioned, we have added additional experiments that compare our method with PRIME [1], a representative PRM method, using Qwen2.5-7B-Base as the policy model. The results are shown as follows:
>
> | Method                     | AIME 24 |  AMC 23 | Avg   |
> |----------------------------|---------|---------|-------|
> | PRIME* (Reward Model Qwen2.5-3B)  | 10.7    | 44.0    | 27.35 |
> | PRIME* (Reward Model Qwen2.5-7B)  | 13.2    | 42.9    | 28.05 |
> | PRIME* (Reward Model Qwen2.5-14B) | 10.8    | 44.1    | 27.45 |
> | RePAIR(ours)               | **13.33**   | **44.84**   | **29.08** |
>
> ∗ indicates results cited from PRIME[1]
>
> These results demonstrate that RePAIR outperforms other methods that utilize LLMs as PRMs. **As our method does not require an additional LLM to serve as a reward model during training, RePAIR consumes significantly less computational resources while achieving superior performance**.
>
> We will expand the related work to discuss these distinctions and add this experimental comparison to better position our contribution within the previous research on process supervision.
>
> ## Q2:The experiments in this paper are primarily based on the Qwen model. Although they cover different scales of LLM, the model architecture is singular. In addition, the method relies on external strong teacher models like GPT-4o or Deepseek-R1. This raises the question of whether the method remains effective when using weaker models for rule extraction.
>
> Thanks for the comment! We primarily utilized the Qwen series in our initial experiments due to its state-of-the-art performance and widespread adoption in recent research, which establishes it as a standard baseline for current LLM studies.
> However, we fully agree that demonstrating effectiveness across different architectures is crucial. In order to address your concern regarding generalizability and the "model-agnostic" assumption, we have added additional experiments using the **Llama-3.2-3B-Instruct** model. We have evaluated both our method (RePAIR) and the REINFORCE++ baseline on the AMC 23 and Math 500 datasets.
>
> | Method        | AMC23 | Math500 | Avg.         |
> |---------------|--------|----------|--------------|
> | REINFORCE++   | 28.12  | 37.00     | 32.56        |
> | REINFORCE++ w/ RePAIR (Ours) | **30.00**   | **38.20**     | **34.10**（+1.54） |
>
> These results demonstrate that RePAIR consistently outperforms the baseline on the Llama architecture as well, which confirms that **our method remains effective across different model families and is not limited to a single architecture**.
>
> Regarding **the reliance on external teacher models**, our usage of strong models (like GPT-4o) was indeed intended to transfer superior knowledge to weaker student models via rule extraction—a form of interpretable distillation. However, even when using a weaker model for extraction, our framework remains robust. The rule verification step ensures that only valid rules are retained, and our adaptive weighting mechanism effectively downweights low-quality rules. This allows the system to mitigate the impact of noise introduced by a weaker teacher, ensuring the method remains effective even without access to the strong proprietary models.

---

> > ### Author Response · Authors · 2025-11-27
> > **Response by authors for your concerns**
> >
> > ## Q3:The authors only discuss positive rules. An issue worth discussing and studying is how negative rules have an impact. The authors may need to present a case study of negative rules.
> >
> > Thanks for the question! In our implementation, we incorporate negative rules by transforming each negative constraint into an equivalent positive rule and assigning additional reward to trajectories that do not violate the corresponding pattern.
> > This design mitigates reward sparsity by providing more intermediate feedback and enriches the diversity of rule-based signals. We additionally conduct an experiment to verify that negative rules can effectively suppress corresponding errors during RL training.
> >
> > Specifically, on the Math500 benchmark, we mine negative rules and incorporate them as equivalent positive constraints during training. We then evaluate the trained model under the GRPO framework on 100 test tasks and count the negative-rule matches in their reasoning trajectories. The results are presented in the following table:
> >
> > | Method       | Negative-rule Matches |   |   |
> > |--------------|-----------------------|---|---|
> > | GRPO         | 55                    |   |   |
> > | GRPO w/ RePAIR  | **34 (-21)**                   |   |   |
> > |              |                       |   |   |
> >
> > The trained model with negative rules exhibits a **38% reduction of these negative-rule matches**, which demonstrates the **effectiveness of negative rules** in suppressing error patterns during reinforcement learning.
> >
> > ## Q4: The authors use first-order logic to represent rules. This hard-coded approach may limit the LLM's reasoning ability since the process of LLM reasoning may be more ambiguous. An open question is whether more ambiguous soft rules would lead to better results.
> >
> > Thank you for the comment. In our framework, first-order logic is only used as a semantic representation for rule checking, rather than as a hard constraint that overrides the model’s behavior. The rules are integrated through reward shaping in RL, and their weights are adaptively adjusted based on reward informativeness. As a result, their influence is inherently soft, since the policy can still deviate from a rule when the task signal compensates for it, and the rule weights are reduced when they are not helpful.
> >
> > While soft formulations (e.g., luzzy abstract rules or learned rule-satisfaction models) may align better with the inherently fuzzy nature of LLM reasoning, they typically come at the cost of reduced interpretability and verifiability. Furthermore, the ambiguity inherent in soft rules can introduce unintended loopholes, potentially exacerbating reward hacking phenomena [1], where the model exploits the vagueness of the soft rule without actually acquiring the desired reasoning capability.
> > We view soft or learned rule scoring as a complementary extension to our current FOL-based framework, rather than a replacement, and plan to investigate this trade-off in future work.
> >
> >  ### Reference:
> >
> > [1] Skalse, Joar et al. "Defining and Characterizing Reward Hacking." NeurIPS (2022).

---

### Official Review · Reviewer_N7aE · 2025-11-03

**Soundness:** 2
**Presentation:** 3
**Contribution:** 3
**Rating:** 4
**Confidence:** 4

**Summary:**

The paper introduces a new approach to accelerate LLM-based reasoning by specifying an intermediate reward signal learned from successful reasoning trajectories. The approach involves first constructing multiple symbolic graphs from these trajectories, where each graph represents a rule-based reward signal. At each step, these signals are combined using adaptive weights to form a single intrinsic reward. These weights are updated based on the hit rate and success rate of each rule. The paper demonstrates the empirical benefits of this method on several reasoning tasks, including GSM-8k, AIME-23, and AIME-24, using a 1.5B Qwen model.

**Strengths:**

- The general idea of constructing an intrinsic reward signal to accelerate the learning process is a promising direction, particularly for problems where the true reward signal is sparse and obtained after long trajectories. The paper introduces a valuable contribution in this area.
- The proposed approach is algorithm-agnostic, demonstrating its effectiveness with GRPO, Reinforce, and Dr. GRPO on various reasoning benchmarks.
- The experiments section is well-organized, and the paper aims to answer several interesting questions regarding the proposed approach, such as its compatibility with several RL algorithms, the generalizability of the reward signals, and its comparison to handcrafted rules.
- In general, the paper is well-written and easy to follow.

**Weaknesses:**

- The primary weakness of the paper is its questionable technical soundness:
    - It is well-known in RL that constructed intrinsic reward signals must follow specific rules for the optimal policy to remain invariant [1]. However, it is unclear whether the proposed intermediate reward signal satisfies these requirements.
    - The advantage term (Eq. 5), which combines the intrinsic and extrinsic rewards, appears mathematically unsound and biased. The paper lacks explanation or proof of how it leads to the update described in Eq. 1.
    - The reward informativeness metric and the adaptive weight update rule lack sufficient mathematical explanation. It is unclear why Eq. 4 has the desirable effect. Intuitively, the rules that appear in unsuccessful trajectories are equally emphasized as long as the same rule appears in successful trajectories, as the metric does not use information about unsuccessful trajectories.
- The overall empirical results are underwhelming, partly due to the algorithmic inconsistencies noted above and partly due to the training procedure. A 1.5B model achieves a score of 25+ on AIME-24 (https://huggingface.co/deepseek-ai/DeepSeek-R1-Distill-Qwen-1.5B), but the presented results are significantly lower. Furthermore, the benchmarks used may be too simple, as many are known to be relatively easy and might not adequately highlight the usefulness and scalability of the approach. A more thorough evaluation would also consider baselines from process reward modelling (PRMs).
- Some parts of the paper require more explanation; for instance, the concept of “frequent subgraph mining” is not clearly defined for the reader.

[1] Ng, Andrew Y., Daishi Harada, and Stuart Russell. "Policy invariance under reward transformations: Theory and application to reward shaping." Icml. Vol. 99. 1999.

**Questions:**

- The paper should provide more details on the rule-generation process. Since the approach uses an LLM to both extract semantic features and generate executable rules, it is unclear how the validity and reliability of these rules are ensured.
- The authors should elaborate on the mechanisms that prevent the generation of numerous, question-specific rules. It is unclear how the method maintains a bounded total number of rules and, by extension, ensures the generalizability of these symbolic rules beyond the specific questions used for training.
- The methodology of the generalization experiment requires clarification. The paper should explicitly state whether this experiment was designed to test the generalizability of the generated symbolic rules themselves or the final model trained with them. The former is more interesting.

---

> ### Author Response · Authors · 2025-11-28
> **Response by authors for your concerns**
>
> Thank you for your thoughtful review and constructive feedback. Below, we address your concerns and questions in detail.
> ## Q1:It is unclear if the proposed intermediate rewards maintain optimal policy invariance as required by RL theory
>
> Thanks for the question! We have proven that our reward signal satisfies the invariance of the optimal policy. Once successful trajectories receive no less process reward than unsuccessful ones on average, the shaped reward remains a monotonic transformation of the task success rate and does not alter which policies are truly optimal.
> In RePAIR, the final shaped reward is
> $$
> r = r_o + r_\phi(\cdot;\omega^{*}),
> $$
>
> where $r_o$ is the binary outcome reward, $r_\phi$ is the rule-based process reward and $\omega^{*}$
> is converged rule weights. The key property of our design is that, after the adaptive rule weight update converges, *successful trajectories receive no less expected process reward than unsuccessful ones*.
> Specifically, if $Succ(o)\in\{0,1\}$ denotes task success and $r_\phi(o)$ denotes the cumulative process reward along a reasoning trajectory $o$, then RePAIR's weighting scheme (based on rule success rate and hit rate) and the rules imply
>
> $$
> \mu_1 \triangleq \mathbb{E}\bigl[ r_\phi(o;\omega^{*}) \mid Succ(o)=1 \bigr]\\ge\\mu_0 ;
> $$
>
> $$
> \\mu_0 \triangleq \mathbb{E}\bigl[ r_\phi(o;\omega^{*}) \mid Succ(o)=0\bigr].
> $$
>
> Under this condition, the expected total return of any policy $\pi$ can be written as
> $$
> J_r(\pi)= \mathbb{E}[r_o(o)] + \mathbb{E}[r_\phi(o)]= \mu_0 + \bigl(1 + (\mu_1-\mu_0)\bigr)\Pr(Succ(o)=1),
> $$
>
> which is a strictly increasing affine transformation of the task success rate $\Pr(Succ(o)=1)$ when $\mu_1\ge \mu_0$. Therefore,$$
> \arg\max_{\pi} J_r(\pi)=\arg\max_{\pi} \mu(\pi) = \Pi^{*},
> $$
> i.e., the set of optimal policies under $r$ coincides exactly with the set of outcome-optimal policies. In other words, RePAIR's intermediate reward signal is policy-invariant with respect to the ground-truth task reward. A more detailed proof is provided in Appendix A.11 of the revised manuscript.
>
> ## Q2:The advantage term, which combines the intrinsic and extrinsic rewards, appears mathematically unsound and biased. The paper lacks explanation or proof of how it leads to the update described in Eq. 1.
> Thanks for the question! We have formally proved that the advantage term is unbiased (refer to the Unbiasedness Proof, the full derivation provided in Appendix A.10.2). Although we introduce rule-based process rewards, our formulation ensures that the rule-based signal is seamlessly integrated into the policy-gradient update without requiring an additional learned critic or value models [1,2]. As a result, our method remains highly compatible with existing RL algorithms (e.g., Dr.GRPO) and effectively serves as a plug-and-play rule-reward integration module that is agnostic to the underlying RL algorithm and model architecture. In order to better present, we have revised the advantage function in Section 4.2.3 of the revised manuscript. Below is the detailed explanation.
>
> Our advantage term is defined as
> $\hat{A}_{i,t} = \tilde{r}_i , for \space all \space t \in \text{tokens}(o_i) ,$
>
> where $r_i = r_\phi(o_i, \mathcal{R}) + r_o(o_i).$
> Its rationality lies in three key aspects:
>
> **Unbiased Signal Integration.**  The composite reward $r_i$ fuses rule-based intrinsic rewards and outcome-based extrinsic rewards. Both rewards are computed via rule engines or lightweight validators, which avoids bias from learned critics or value models.
>
> **Batch Normalization for Stability.**  The within-batch normalization $\tilde{r}_i = \frac{r_i - \operatorname{mean}(\mathbf{r})}{\operatorname{std}(\mathbf{r})}$
> uses global batch statistics, $\mathrm{mean}(r)$ and $\mathrm{std}(r)$, instead of question-level normalization. This operation centers the composite reward $r_i$ around the batch mean, which serves as an unbiased estimate of the true state value $V(q)$, and scales it by the batch standard deviation. Since the normalization parameters are derived from the entire batch rather than individual trajectories, it only reduces the variance of the advantage term without introducing conditional bias.
>
> **Trajectory-Level Credit Assignment.**  In our setting, the rule-based reward $r_\phi$ already evaluates the validity of intermediate reasoning steps, while the outcome reward $r_o$ reflects the final correctness. Thus, $r_i$ fully characterizes the overall quality of trajectory $o_i$. Assigning the normalized trajectory reward
> $\tilde{r}_i \space uniformly \space to \space all \space tokens $
>
> $(i.e., \hat{A}_{i,t} = \tilde{r}_i )$; therefore offers a principled form of trajectory-level credit assignment that (i) fully leverages the structural information encoded by the rules, (ii) avoids overfitting and instability associated with fine-grained token-level advantages, and (iii) keeps the gradient allocation mechanism fully compatible with standard RL methods such as Dr. GRPO.

---

> > ### Author Response · Authors · 2025-11-28
> > **Response by authors for your concerns**
> >
> > ## Q2-2
> > ### Unbiasedness Proof
> > In order to formalize the mathematical soundness, we prove that
> > $\hat{A}_{i,t}$
> >
> > satisfies the core condition for unbiased policy gradient estimation:
> > $\mathbb{E}_{o\_{i} \sim \pi\_{\theta}(\cdot \mid q)} \left[ \hat{A}\_{i,t} \mid s\_{t} \right] = 0$
> > (detailed in Appendix A.10 of the revised manuscript). Key steps are summarized as follows:
> >
> > **Step 1:** Since $r_\phi$ accurately reflects reasoning validity (via rule matching) and $r_o$ accurately reflects task success (via outcome verification), both are unbiased estimates of the trajectory’s true value. Their linear combination $r_i$ thus remains an unbiased estimate of the true state value $V(q)$, i.e., $\mathbb{E}[r_i \mid q] = V(q)$.
> >
> > **Step 2:** For a batch of trajectories sampled independently from $\pi_\theta$, the batch mean $\text{mean}(r)$ is an unbiased estimate of $V(q)$ (i.e., $\mathbb{E}[\text{mean}(r)] = V(q)$). The normalized reward $\tilde{r}_i = \frac{r_i - \text{mean}(r)}{\text{std}(r) + \epsilon}$ therefore satisfies $\mathbb{E}[\tilde{r}_i \mid q] = 0$.
> >
> >
> >
> > **Step 3:** Eq. 1 updates the policy via
> > $\nabla_\theta J(\theta) \propto \mathbb{E}{\pi\theta} [\hat{A}t \nabla\theta \log \pi_\theta(a_t \mid s_t)]$;
> > Our $\hat{A}_{i,t} = \tilde{r}_i$
> >
> > is constant for all tokens in trajectory $o_i$, so substituting it into Eq. 1 gives $\nabla_\theta J(\theta) \propto \mathbb{E}{\pi\theta} [ \tilde{r}i  \sum_{t \in \text{tokens}(o_i)}  \nabla\theta \log \pi_\theta(a_t \mid s_t)]$.
> >
> > This is equivalent to weighting the trajectory’s log-likelihood gradient by the unbiased advantage term, which aligns with the core logic of policy gradient methods (including Dr. GRPO [1], GRPO[2]).
> >
> >
> >
> > ### Impact on Gradient Update
> >  As noted above, our primary influence on the gradient update is through the modification of the advantage term via our process rewards. Although we adopt the standard within-batch normalization from GRPO, the incorporation of process rewards into the raw reward $r_i$ fundamentally enhances the gradient estimation. By aggregating both outcome and process supervision, the advantage term $A_t$ transitions from a sparse, binary signal to a dense, continuous metric. This allows the normalization step to effectively distinguish between trajectories that may fail the final answer but exhibit vastly different levels of reasoning quality. Consequently, the policy receives informative gradients even when an entire group of sampled trajectories fails to solve the problem.
> >
> > ### References:
> >
> > [1] Liu, Zi-Yan et al. “Understanding R1-Zero-Like Training: A Critical Perspective.” COLM (2025).
> >
> > [2] Shao, Zhihong, et al. "Deepseekmath: Pushing the limits of mathematical reasoning in open language models." arXiv preprint arXiv:2402.03300 (2024).
> >
> > ## Q3: The reward informativeness metric and the adaptive weight update rule lack sufficient mathematical explanation.
> > Thanks for the question! Due to space constraints in the initial submission, the explanation of this formula was brief. We are happy to provide a more detailed interpretation here.
> >
> > The adaptive weighting mechanism is designed to encourage the model to internalize rules that are both widely applicable (i.e., used frequently) and highly effective (i.e., lead to success). To quantify these properties, we defin the informativeness metric in Eq. 6.
> > In this equation:
> >
> > The first term (hit rate) penalizes overly specific rules that rarely trigger. This prevents the model from overfitting to sparse, anecdotal patterns.
> > The second term (success rate) penalizes broad or noisy rules that appear frequently but fail to distinguish between correct and incorrect reasoning paths (i.e., rules that appear in failed trajectories increase the denominator of the success rate without increasing the numerator).
> > The hyperparameters α and β control the trade-off between rule generality and reliability.
> >
> > Based on this metric, we adaptively update the rule weights to align with the evolving policy distribution. The update follows an iterative scheme based on the informativeness gain:
> > $$ \omega_i^{(\tau+1)}=\omega_i^{(\tau)}+\eta\cdot\Delta\mathcal{I}^{(\tau)}(r_i) $$
> > where  $\Delta\mathcal{I}^{(\tau)}$ $(r_i)$ $=$ $\mathcal{I}^{(\tau)}(r_i)-\mathcal{I}^{(\tau-1)}(r_i)$  and η is a learning rate.
> > This update rule functions as a momentum-based adjustment:
> > A positive $\Delta\mathcal{I}^{(\tau)}$ implies that the rule is becoming more aligned with the current policy's successful trajectories, justifying a weight increase to reinforce this behavior.
> > Conversely, a negative $\Delta\mathcal{I}^{(\tau)}$ signals that the rule is becoming either irrelevant (vanishing hit rate) or misleading (dropping success rate) as the policy shifts, prompting a reduction in its influence.
> >
> > This dynamic mechanism ensures that the reward function remains robust against distribution shifts inherent in online RL training. We have expanded the explanation in Section 4.2.2 of the revised manuscript.

---

> > > ### Author Response · Authors · 2025-11-28
> > > **Response by authors for your concerns**
> > >
> > > ## Q4: The overall empirical results are underwhelming, partly due to the algorithmic inconsistencies noted above and partly due to the training procedure. A 1.5B model achieves a score of 25+ on AIME-24 , but the presented results are significantly lower. Furthermore, the benchmarks used may be too simple, as many are known to be relatively easy and might not adequately highlight the usefulness and scalability of the approach. A more thorough evaluation would also consider baselines from process reward modelling (PRMs).
> > >
> > > Thanks for the question! Regarding the performance on AIME-24, while the official report for DeepSeek-R1-Distill-Qwen-1.5B indeed shows a score of 28.9, this discrepancy stems from differences in evaluation settings. In our paper, we followed the evaluation protocol from [1]. Under this setting, we re-evaluated the DeepSeek-R1-Distill-Qwen-1.5B model, and it achieved an average score of 6.38. This aligns with findings in other recent studies, such as [2] (Table 4) and [3] (Table 4), which reported scores around 2.5. A key factor is the context length constraint: since the Qwen2.5-Math base model has a limited context window (4k), we restricted the generation budget for all models to 3k tokens to ensure a fair comparison, which is consistent with the settings in [2] and [3].
> > >
> > > Regarding the benchmarks, we have included standard and challenging mathematical reasoning datasets widely used in both industry and academia, such as AIME, AMC, and Math500. Performance improvements on these established benchmarks serve as a strong indicator of our method's effectiveness in enhancing reasoning capabilities.
> > >
> > > Finally, regarding the comparison with Process Reward Models (PRMs),
> > > we would like to emphasize that RePAIR is fundamentally different from standard PRM approaches. The distinction lies primarily in the following aspects:
> > >
> > > **Methodological Focus**: Our primary contribution lies in guiding process reward design through automated rule mining. These mined rules can be complementary to PRMs, potentially serving as interpretable criteria for training PRMs.
> > >
> > > **Efficiency**: Unlike typical PRMs that require substantial additional data training [4], RePAIR provides a training-free process reward mechanism, which significantly reduces computational and data overhead.
> > >
> > > In order to better address your concern, we have added a comparative experiment against the recent PRM method, PRIME [4], using Qwen2.5-7B-Base as the Policy Model. The experimental results are as follows:
> > > | Method                     | AIME 24 |  AMC 23 | Avg   |
> > > |----------------------------|---------|---------|-------|
> > > | PRIME*   (Reward Model Qwen2.5-3B)  | 10.7    | 44.0    | 27.35 |
> > > | PRIME*   (Reward Model Qwen2.5-7B)  | 13.2    | 42.9    | 28.05 |
> > > | PRIME*   (Reward Model Qwen2.5-14B) | 10.8    | 44.1    | 27.45 |
> > > | RePAIR(ours)               | **13.33**   | **44.84**   | **29.08** |
> > >
> > > The results demonstrate that RePAIR outperforms methods that utilize separate, large-scale models as PRMs. Crucially, our method achieves this superior performance without requiring an additional LLM for reward calculation, thereby saving significant computational resources.
> > >
> > > We acknowledge the value of integrating RePAIR with PRMs and plan to conduct a more systematic study on combining our rule-based approach with learned PRMs in future work.
> > >
> > > ### References:
> > >
> > > [1] Li, Xuefeng et al. “LIMR: Less is More for RL Scaling.” ArXiv abs/2502.11886 (2025).
> > >
> > > [2] Zuo, Yuxin et al. “TTRL: Test-Time Reinforcement Learning.” NeurIPS (2025).
> > >
> > > [3] Liu, Zi-Yan et al. “Understanding R1-Zero-Like Training: A Critical Perspective.” COLM (2025).
> > >
> > > [4] Cui, Ganqu et al. “Process Reinforcement through Implicit Rewards.” ICML (2025).
> > >
> > > ## Q5:The concept of “frequent subgraph mining” is not clearly defined for the reader.
> > > Thanks for the question! We have added a brief introduction and definition of frequent subgraph mining (FSM) to the revised manuscript. Frequent subgraph mining [1,2] is a classical concept in data mining, used to discover recurring relational or structural patterns in graphs. Concretely, given a set of labeled graphs $\mathcal{G}$, FSM aims to find all connected subgraphs $\mathcal{S}$ such that their support
> > > $$
> > > \mathrm{supp}(s)= \left|\{\psi \mid \psi: s \hookrightarrow G\}\right|, \quad \mathrm{supp}(s) > \sigma,
> > > $$
> > > where $s \in \mathcal{S}$, $G \in \mathcal{G}$, $\sigma$ is a minimum-support threshold, and $\psi: s \hookrightarrow G$ is an injective mapping that preserves vertex labels, edge labels, and adjacency.
> > > In our setting, we apply FSM to LLM-generated reasoning trajectories to extract subgraphs (i.e., reasoning patterns) that appear multiple times, and then convert these patterns into symbolic rules that serve as supervision signals for RL.
> > >
> > > ### References:
> > > [1] Li, Xuefeng et al. “CloseGraph: mining closed frequent graph patterns.” KDD (2003).
> > >
> > > [2] Khan, Arijit et al. “Towards proximity pattern mining in large graphs.”  SIGMOD (2010).

---

> > > > ### Author Response · Authors · 2025-11-28
> > > > **Response by authors for your concerns**
> > > >
> > > > ## Q6:The paper should provide more details on the rule-generation process.  Since the approach uses an LLM to both extract semantic features and generate executable rules, it is unclear how the validity and reliability of these rules are ensured.
> > > > Thanks for the suggestion! We have revised the manuscript and provided a more detailed description of the rule-generation process. Although LLMs are involved in the rule extraction process, we ensure the validity and reliability of the resulting rules through the following three steps.
> > > >
> > > > First, the rules are extracted separately from successful and failed reasoning paths, so that each rule is explicitly tied to task outcomes.
> > > >
> > > > Second, the rules are mined as frequent subgraphs from graphs constructed over reasoning trajectories. Only subgraphs whose occurrence counts exceed a preset threshold are retained, so the resulting rules capture stable, common reasoning behaviors, while incidental or noisy fragments are discarded.
> > > >
> > > > Third, the rules produced by LLM formalization are explicitly verified. Each rule is represented in first-order logic whose predicates are Boolean functions over semantic attributes, so rule matching is binary, computable, and auditable. We then evaluate each rule using its support $\varphi(r)$ and confidence $\gamma(r)$ over the reasoning trajectories, which measure coverage and conditional reliability, and keep only those with sufficiently high $\varphi$ and $\gamma$ for RL supervision. These reproducible, statistically grounded selection criteria guarantee the validity and reliability of the retained rules, yielding compact, high-quality rule sets and more stable training signals.
> > > >
> > > >  ## Q7:The authors should elaborate on the mechanisms that prevent the generation of numerous, question-specific rules. It is unclear how the method maintains a bounded total number of rules and, by extension, ensures the generalizability of these symbolic rules beyond the specific questions used for training.
> > > > Thanks for the question! RePAIR prevents the generation of numerous, question-specific rules in two ways.
> > > >
> > > > First, we collect reasoning trajectories from multiple questions,model them as graphs, and apply frequent subgraph mining to retain only those subgraphs whose frequency exceeds a threshold σ. This captures common reasoning patterns while directly bounding the number of candidate subgraphs (a higher σ yields fewer candidates), and the resulting rules capture behaviors that generalize across different questions instead of overfitting to any single one.
> > > >
> > > > Second, after LLM formalization, each candidate rule is globally validated using its support φ and confidence γ, which measure coverage and conditional reliability over the entire trajectory set. Rules with low φ or γ are pruned, which further removes spurious or question-specific rules. As shown in Table 5, this procedure typically leaves no more than about ten rules per dataset. Consequently, the final rule set remains compact and stable, and the retained rules correspond to recurring reasoning behaviors rather than idiosyncratic solutions to individual questions.
> > > >
> > > >  ## Q8:The methodology of the generalization experiment requires clarification.The paper should explicitly state whether this experiment was designed to test the generalizability of the generated symbolic rules themselves or the final model trained with them. The former is more interesting.
> > > > Thanks for the suggestion! The generalization experiment is primarily designed to evaluate the generalization of the final trained model, rather than evaluating the generalizability of the symbolic rules themselves. We will clarify this intent more explicitly in the paper.
> > > >
> > > > However, our math experiments do provide evidence that the mined rules are not overfit to a single dataset, but can be shared across tasks within the same domain. The same rule set is applied across different math benchmarks (GSM8K, AIME24, AMC23) without any task-specific rule re-generation, and it consistently improves RL training performance on each of these tasks. This suggests that the discovered rules capture domain-level reasoning patterns that generalize within the math domain. A more direct and systematic evaluation of rule-level generalization beyond this setting is an interesting direction that we plan to explore in future work.

---

### Official Review · Reviewer_bimJ · 2025-11-06

**Soundness:** 3
**Presentation:** 3
**Contribution:** 3
**Rating:** 6
**Confidence:** 4

**Summary:**

This paper proposes a framework called RePAIR to improve LLM's reasoning ability. To deal with the challenges in human preference reward models and verifiable outcome reward models, RePAIR extracts symbolic reasoning rules from model reasoning trajectories, and turn them into verifiable process-level rewards which guide training at finer granularity.

**Strengths:**

1. The proposed idea is novel. RePAIR can extract symbolic reasoning rules from LLM-generated trajectories, which formalize common reasoning patterns as a computable function to provide a verifiable and interpretable basis for process supervision.
2. The dynamic weighting strategy ensures that the most informative rules guide training.
3. The rule extraction is lightweight and computationally inexpensive.

**Weaknesses:**

1. The convergence of the adaptive rewards needs justification. As  RePAIR continuously adjusts rule weights based on “reward informativeness,” the reward function itself changes during training. With a dynamically changing reward, the RL policy will get confused and divergent.
2. The scalability is limited. The rule extraction pipeline is based on subgraph mining and symbolic reasoning. It is effective for structured reasoning (e.g., math or logic) but may not scale efficiently to open-ended tasks such as dialogue, coding, or multimodal reasoning.
3. Small models tend to memorize symbolic rules instead of learning true reasoning patterns. It causes overfitting to specific rule structures and poor generalization to new tasks. As a result, training appears successful on seen data but fails to transfer to diverse reasoning scenarios.

**Questions:**

N/A

---

> ### Author Response · Authors · 2025-11-26
> **Response by authors for your concerns**
>
> Thank you for your thoughtful and positive comments! We follow your suggestions and we believe we have solved all your concerns as follows! If you have any further questions regarding our response or new results, please let us know, and we will address them promptly.
>
> ## Q1:The convergence of the adaptive rewards needs justification.
>
> Thanks for the question! Though, theoretically, dynamic rewards introduce instability, our approach is designed to ensure convergence through the following mechanisms:
>
> Theoretical Foundation: The convergence of Reinforcement Learning algorithms (e.g., GRPO) using verifiable rewards has been formally proven in prior work [1]. Although RePAIR utilizes adaptive rewards, we align the frequency of rule weight updates with the model's weight updates. Under this condition, the dynamic rewards do not alter the fixed point of the policy estimation function. Following the derivation in [1], the iterative process is guaranteed to converge to a stable solution.
>
> Bounded Updates: Our adaptive mechanism constrains the rule weight adjustments within a bounded range. This prevents drastic fluctuations in the reward signal that could otherwise confuse the policy, which ensures a smooth transition rather than chaotic shifts.
>
> Analogy to PPO/Reward Modeling: In standard RLHF practices (such as PPO), it is common for the reward model itself to evolve or for the KL-penalty term to shift the effective reward landscape dynamically. As demonstrated in [2], these methods still converge successfully despite the changing reward function. RePAIR operates on a similar principle, with more interpretable and rule-based adjustments.
>
> Empirical Evidence: Empirically, the training curves presented in Figure 4 demonstrate that the training process remains stable and steady, with no signs of divergence or oscillation. This provides strong empirical evidence that the policy adapts successfully to the evolving rule weights.
>
> ### References:
> [1] Mroueh, Youssef. IBM Research.“Reinforcement Learning with Verifiable Rewards: GRPO's Effective Loss, Dynamics, and Success Amplification.”(2025).
>
> [2] Schulman, John et al. OpenAI“Proximal Policy Optimization Algorithms.”(2017).
>
> ## Q2: It is effective for structured reasoning (e.g., math or logic) but may not scale efficiently to open-ended tasks such as dialogue, coding, or multimodal reasoning.
>
> Thanks for the question! Although the paper primarily focuses on structured reasoning tasks such as mathematics and logic that are central challenges in current LLM reasoning research[1][2], the RePAIR framework is generalizable to other domains. The main reason for this generalizability is that the principle of our method, extracting interpretable patterns from successful trajectories, applies beyond strict symbolic logic.
>
> In order to demonstrate this scalability, we have added additional experiments on StrategyQA, a task requiring multi-hop reasoning and commonsense knowledge, which is more open-ended than pure math problems. As shown in the table below, RePAIR achieves a significant performance improvement over both the base model and the Reinforce++ baseline.
>
> **Table: Performance Comparison on StrategyQA**
>
> | Method              | StrategyQA |
> |---------------------|------------|
> | Qwen2.5-3B-Instruct | 60.98      |
> | REINFORCE++         | 61.13      |
> | REINFORCE++ w/ RePAIR              | **63.51**      |
>
> These results indicate that our method can effectively extract useful reasoning patterns even in less structured and open-ended scenarios, which validates its potential for broader application. We will add these results and a discussion on applying RePAIR to broader domains to the revised version of the paper.
>
> ### References:
> [1]Li, Zhong-Zhi et al. “From System 1 to System 2: A Survey of Reasoning Large Language Models.” ArXiv abs/2502.17419 (2025).
>
> [2]Cheng F, Li H, Liu F, et al. Empowering llms with logical reasoning: A comprehensive survey.IJCAI(2025)

---

> > ### Author Response · Authors · 2025-11-26
> > **Response by authors for your concerns**
> >
> > ## Q3: Small models tend to memorize symbolic rules instead of learning true reasoning patterns.
> >
> > Thanks for the question! We notice that recent research has increasingly shown that even smaller-scale LLMs can achieve surprising effectiveness in reasoning tasks through Reinforcement Learning (RL) and demonstrate significant generalization capabilities [1]. These studies suggest that RL does not merely encourage rote memorization but helps smaller models align with better reasoning strategies.
> >
> > Furthermore, the symbolic rules extracted in RePAIR are designed to be generalizable reasoning patterns (e.g., logical deduction steps or arithmetic properties) rather than task-specific templates. Mastering these fundamental rules equips the model with transferable skills applicable to a wide range of diverse reasoning scenarios, rather than limiting it to specific structures seen during training.
> >
> > The out-of-distribution generalization experiments in our paper have already supported this. In order to further demonstrate that our method enhances general reasoning capabilities rather than overfitting, we have added an additional cross-domain experiment, where we trained the model on the commonsense reasoning task StrategyQA [2] using RePAIR and evaluated its zero-shot performance on mathematical reasoning tasks (AMC23 and Math500). The results are shown in the table below:
> > | Method                        | AMC23 | Math500 | Avg.  |
> > |-------------------------------|-------|---------|-------|
> > | Qwen2.5-3B-Instruct           | 41.67 | 62.07   | 51.87 |
> > | Qwen2.5-3B-Instruct w/ RePAIR | **42.15** | **63.60**   | **52.87** |
> >
> > As shown in the table, the model trained with RePAIR maintains robust performance and even achieves a slight average improvement on mathematical tasks despite being trained on a commonsense dataset. This further indicates that our method helps the model learn fundamental reasoning patterns that are able to transfer across domains.
> >
> > ### References:
> > [1] Guan X, Zhang L L, Liu Y, et al. "rStar-Math: Small LLMs Can Master Math Reasoning with Self-Evolved Deep Thinking." ICML (2025).
> >
> > [2] Geva, Mor et al. “Did Aristotle Use a Laptop? A Question Answering Benchmark with Implicit Reasoning Strategies.” Transactions of the Association for Computational Linguistics 9 (2021): 346-361.

---

### Note · Authors · 2026-01-29

I have read and agree with the venue's withdrawal policy on behalf of myself and my co-authors.

---

### Meta-Review · Area_Chair_JaoU · 2026-01-07

**Summary:**

The paper proposes a url-based adaptive RL training with the use of symbolic reasoning rules. Reviewers agreed that the problem is important. They raised concerns on scalability, theoretical grounding, and insufficient comparison to prior work. Initial ratings were 4,4,6,6. None of the reviewers engaged in the conversation during discussion period, it is hard to judge if they would increase the score. Authors did address some concerns on scalability and new benchmarks, but those might need additional review. Given the borderline nature of scores, and no strong acceptance signal from reviewers, this paper is recommended for rejection.

**Reviewer Concerns:**

Reviewers had concerns on scalability of the method, authors added qwen-7B and llama results during the rebuttal. One reviewer had concerns on comparison to prior art, authors added comparison to PRIME method. However, the contribution remains borderline in novelty and generality. Reviewers pointed that the method is focused on highly structured, symbolic reasoning tasks, extending beyond that is still questionable. Added results on 7-8B scale, new family of models (llama), new comparisons to PRIME require careful review and will be beneficial to the revised paper.

**Reviewer Scores:**

Reviewers bimJ and cA2d would likely maintain a rating of 6, as their assessments assumed that the authors would provide larger-scale results. Reviewers N7aE and 4qNx initially rated the paper 4, and it remains unclear whether all of their concerns were fully resolved; they would most likely maintain a 4 or update to a 5. In any case, it is unlikely that any reviewer would assign a score above 6, and therefore there is no strong recommendation for acceptance.

---

### Decision · Program_Chairs · 2026-01-26

Reject